# A facile strategy for realizing room temperature phosphorescence and single molecule white light emission

Jianguo Wang[1,2], Xinggui Gu[1,3], Huili Ma[4], Qian Peng[5], Xiaobo Huang[6], Xiaoyan Zheng[1], Simon H.P. Sung[1], Guogang Shan[1], Jacky W.Y. Lam[1], Zhigang Shuai[4] & Ben Zhong Tang [1,7,8]

Research on materials with pure organic room temperature phosphorescence (RTP) and their application as organic single-molecule white light emitters is a hot area and relies on the design of highly efficient pure organic RTP luminogens. Herein, a facile strategy of heavy-atom-participated anion–$\pi^+$ interactions is proposed to construct RTP-active organic salt compounds (1,2,3,4-tetraphenyloxazoliums with different counterions). Those compounds with heavy-atom counterions (bromide and iodide ions) exhibit outstanding RTP due to the external heavy atom effect via anion–$\pi^+$ interactions, evidently supported by the single-crystal X-ray diffraction analysis and theoretical calculation. Their single-molecule white light emission is realized by tuning the degree of crystallization. Such white light emission also performs well in polymer matrices and their use in 3D printing is demonstrated by white light lampshades.

[1] Department of Chemistry, Hong Kong Branch of Chinese National Engineering, Research Center for Tissue Restoration and Reconstruction, Division of Life Science Institute of Advanced study, and Division of Biomedical Engineering, The Hong Kong University of Science and Technology, Clear Water Bay, Kowloon, Hong Kong, China. [2] Key Laboratory of Organo-Pharmaceutical Chemistry, Gannan Normal University, Ganzhou 341000, China. [3] Beijing Advanced Innovation Center for Soft Matter Science and Engineering, Beijing University of Chemical Technology, Beijing 100029, China. [4] Key Laboratory of Organic Optoelectronics and Molecular Engineering, Department of Chemistry, Tsinghua University, Beijing 100084, China. [5] Key Laboratory of Organic Solids, Beijing National Laboratory for Molecular Science, Institute of Chemistry, Chinese Academy of Sciences, Beijing 100190, China. [6] College of Chemistry and Materials Engineering, Wenzhou University, Wenzhou 325035, China. [7] NSFC Center for Luminescence from Molecular Aggregates, SCUT-HKUST Joint Research Institute, State Key Laboratory of Luminescent Materials and Devices, South China University of Technology, Guangzhou 510640, China. [8] HKUST-Shenzhen Research Institute, No. 9 Yuexing 1st RD, South Area, Hi-tech Park, Nanshan, Shenzhen 518057, China. Correspondence and requests for materials should be addressed to X.G. (email: guxinggui@mail.buct.edu.cn) or to B.Z.T. (email: tangbenz@ust.hk)

Materials with room temperature phosphorescence (RTP) have attracted great interest in recent years because of the involved scientific value and practical application[1–6]. Luminogens with RTP features are typically inorganics or organometallic complexes. They have been widely used in biomedicine[7,8], anti-counterfeiting[9], sensors[10], and organic light-emitting diodes[11–13], and exhibit superior performance than their fluorescent counterparts. However, they more or less suffered from poor processability, high cost, and heavy-metal toxicity[13]. Recently, pure organic RTP phosphors have been attracted much attention as advanced phosphorescent luminogens and possess good processability, low cost, versatile molecular design, and facile functionalization[14,15]. The sufficient utilization of the triplet state energy[16,17] enables them to be applied in white light illumination with advantages of high energy conversion efficiency and low heat value. These meet the requirements of energy conservation[18–21]. In particular, organic single-molecule white light emitters (OSMWLEs) based on pure organic RTP have been increasingly pursued by researchers due to their superior performance such as no phase segregation and color aging, good reproducibility, improved stability, and simple device fabrication[22–24]. Although recent research demonstrated that highly efficient OSMWLEs could be realized by the combination of fluorescent and phosphorescent or two different phosphorescent emission of pure organic RTP luminophores[25,26], the research on this field is still in its infancy and remains challenging due to the lack of excellent pure organic RTP luminogens[27]. Therefore, it is of great significance to develop pure organic RTP luminogens with two (blue and yellow) or three (blue, green, and red) complementary emission colors for OSMWLEs.

The current popular strategies to achieve pure organic RTP luminogens are to introduce heavy atoms, heteroatoms (N, O, S, P, and so on) or charge transfer state into the luminescent skeletons to facilitate the effective intersystem crossing (ISC)[28–31] and to modulate the aggregation behaviors to suppress the non-radiative dissipation by polymer aggregation[32], crystallization[33–36], or supramolecular assembly[37–39]. Up to now, some pure organic RTP luminogens have been developed involving mainly keto[26,31,40], carbazole[17,33,41,42] and borate[43–46] functionalities, and their RTP properties were generally tuned by heavy atom effect because of the efficient enhancement of spin–orbit coupling (SOC) to promote the ISC process. Among them, external heavy atom effect (EHE) has been extensively studied since the early 1950s[47]. Many efforts have been devoted to understand the nature of EHE because it is of essential importance to RTP in molecular design[48–50]. Although the EHE has routinely served as an effective strategy to promote the occurrence of phosphorescence[51], the driving force of interactions, especially non-covalent interactions such as ion–π interactions, between the external heavy atom and the luminophore has not been systematically studied. Therefore, in-depth study of the non-covalent interactions is of special interest to explore effective approaches to exert EHE on the design of RTP luminogens with aggregation-induced emission (AIE) feature. Previously, a series of organic salts with AIE features have been developed by our group based on a facile strategy of anion–π+ interactions[52–56]. The specific positions of counterions on both sides of positively charged aromatic ring facilitate the strong anion–π+ interactions[57–59], which block the intermolecular π–π stacking efficiently and induce strong fluorescence of the molecules in the aggregated state. If the counterions are exchanged by heavy halide ions such as iodide (I⁻) and bromide ion (Br⁻), RTP-active organic salts are envisioned to be prepared based on the EHE via anion–π+ interactions, opening a new avenue to design pure organic RTP luminogens. Moreover, the emission of these molecules would be expected to be manipulated to the desired applicaion in OSMWLEs.

With this in mind, in this contribution, we show the construction of RTP luminophores by simple exchange of the counterion of an organic salt, namely, 1,2,3,4-tetra-phenyloxazolium hexafluorophosphate (TPO-P, Fig. 1a)[52] with strong blue emission by heavy halide ions. Due to the anion–π+ interactions between the heavy halide ions and the positively charged aromatic ring, the external heavy halide ions are drawn to be closed to the core chromophore to enhance the SOC to

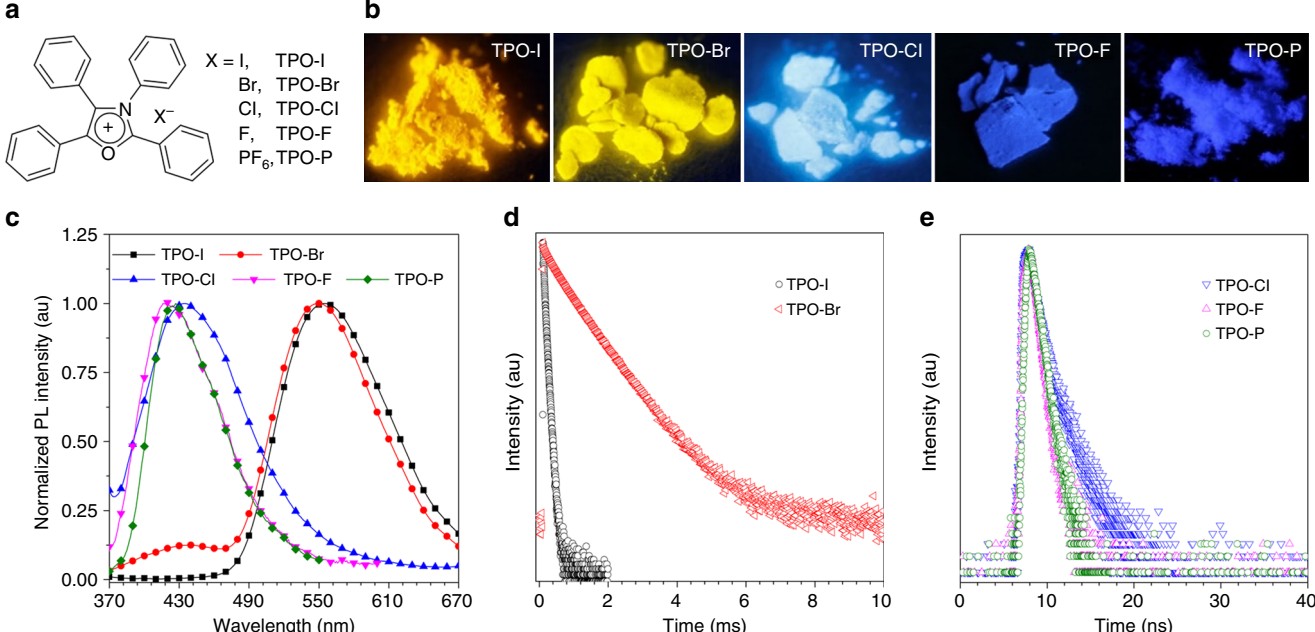

**Fig. 1** Molecule design and photophysical properties. **a** Chemical structures of TPO-I, TPO-Br, TPO-Cl, TPO-F, and TPO-P. **b** Luminescent photographs and **c** PL spectra of TPO-I, TPO-Br, TPO-Cl, TPO-F, and TPO-P in the solid state. Luminescent photographs were taken under 365 nm UV irradiation. Time-resolved PL decay of **d** TPO-I (@ 559 nm) and TPO-Br (@ 549 nm) and **e** TPO-Cl (@435 nm), TPO-F (@ 420 nm), and TPO-P (@ 422 nm) in the solid state at room temperature under air

boost the ISC process effectively. Such kind of interactions is defined as heavy-atom-participated anion–$\pi^+$ interactions, which could serve as a facile strategy to design RTP luminogens. Guided by this strategy, heavy halide ions of $I^-$ and $Br^-$ are introduced to TPO as its counterions to generate TPO-I and TPO-Br. Yellow RTP emission of TPO-I and TPO-Br with the characteristic of long lifetime up to millisecond in the solid state is observed. In contrast, the emission of TPO-P, TPO-Cl, and TPO-F is fluorescence in nature as the hexafluorophosphate ($PF_6^-$), chloridion ($Cl^-$) and fluorinion ($F^-$) showed negligible heavy atom effect. Analysis by single-crystal X-ray diffraction and theoretical calculation further verify the crucial role of heavy-atom-participated anion–$\pi^+$ interactions on efficient RTP. Besides the yellow RTP, TPO-Br also exhibit blue fluorescence in the solid state. The ratio of intensity between phosphorescence and fluorescence could be easily manipulated by the degree of crystallization to meet the requirement of white light for OSMWLEs. In addition, the OSMWLEs could also be fabricated by smart controlling the ratios of counterions in organic salts of TPO-I/Cl or TPO-I/P. All these OSMWLEs also perform well in polystyrene (PS) or poly-ethylene glycol (PEG) matrices. As proof-of-concept, TPO-Br could serve as an excellent additive in PEG for three-dimensional (3D) printing of white light lampshades. Collectively, this work provides a promising strategy of heavy-atom-participated anion–$\pi^+$ interactions to construct RTP-active orangic salts, which greatly extends the design rationales for RTP phosphors and will motivate scientists to develop more advanced RTP luminophores for various high-tech applications such as white light display and 3D printing.

## Results

**Synthesis and photophysical properties**. First of all, organic salts based on TPO with different counterions of $I^-$, $Br^-$, $Cl^-$, $F^-$, and $PF_6^-$ were prepared as shown in Fig. 1a. Among them TPO-P was obtained according to our previous work[52], and the others such as TPO-I, TPO-Br, TPO-Cl, and TPO-F were simply synthesized via cyclization reaction of compound **3** upon phosphorus pentoxide and subsequent ion exchange with sodium halide (NaX) (Supplementary Figure 1). All the structures of these products were characterized by NMR and high-resolution mass spectrometry (HRMS) with satisfactory results (Supplementary Fig. 2-13).

All organic salts have the same absorption peaks at 317 nm accompanying with the weak fluorescence around 440 nm in ethanol solution (Supplementary Fig. 14, 15 and Table 1). In powder, TPO-I and TPO-Br presented yellow emission upon 365 nm UV lamp, while TPO-Cl and TPO-F displayed blue emission similar to the reported TPO-P (Fig. 1b). All of them exhibited the

corresponding higher quantum yields of 16.00, 17.85, 20.05, 11.11, and 18.58% than those of about 1% in solution, indicative of the typical AIE characteristic (Table 1). In addition, the microcrystalline of TPO-I and TPO-Br showed stronger emission with the quantum yields of 35.00% and 36.56% which are nearly two times as large as those of their powders. Similar emission enhancement in the crystalline state were also reported by others[60–62]. The steady-state photoluminescence (PL) spectra of five organic salts in powder were further investigated (Fig. 1c). TPO-Cl, TPO-F, and TPO-P emitted at 435, 420, and 422 nm, while TPO-I and TPO-Br exhibited the exceptional yellow emission at 559 and 549 nm. Notably, TPO-Br also had a small shoulder emission band at 434 nm. The question why TPO-I and TPO-Br exhibited the different emission properties motivated us to get insight into the origin of emission.

The time-resolved PL decay curves of five organic salts were recorded. In solution, their lifetimes at their maximum emission wavelengths were evaluated to be nanosecond scale (Supplementary Fig. 16, Supplementary Table 1, and Table 1), definitely belonged to fluorescence. Similarly, the blue emission of TPO-Br, TPO-Cl, TPO-F, and TPO-P powders was also confirmed to be fluorescence because of the short lifetimes in nanosecond (Fig. 1e, Table 1, and Supplementary Table 2). Interestingly, TPO-I and TPO-Br with the yellow emission at 559 and 549 nm possessed the long lifetimes with the average values of 48 μs and 706 μs, respectively (Fig. 1d and Table 1). The delayed PL spectra collected by delay of 50 μs further demonstrated that the yellow emission of TPO-I and TPO-Br is assigned to the long-lived one (Supplementary Fig. 17). Under 77 K, the time-resolved PL decay curves of TPO-I and TPO-Br were measured (Supplementary Fig. 18) and their lifetimes were largely increased compared to those at room temperature, with the average lifetimes up to 88 μs and 1.65 ms, respectively (Supplementary Table 3), which rule out the thermally activated delayed fluorescence[63]. Thus, the yellow emission of TPO-I and TPO-Br is identified to be RTP. In addition, the lifetimes of TPO-I and TPO-Br at 559 nm and 549 nm in vacuum were also evaluated with the average values of 52 μs and 763 μs, respectively (Supplementary Fig. 19 and Supplementary Table 4), which were slightly larger than that in air, suggesting that such phosphorescence lifetimes of TPO-I and TPO-Br in the solid state were negligibly sensitive to oxygen. All RTP decays can be fitted with two exponents for TPO-I and three exponents for TPO-Br, which is possibly due to the formation of various aggregates in the solid state. Further, we continued to investigate the photophysical properties of TPO-I and TPO-Br under different conditions such as dielectric constants of solvents, concentration, and temperature. As shown in Supplementary

| AIEgens | $\lambda_{abs}$ (nm) | $\lambda_{em}$ (nm) | | $\tau$ (ns)[a] | | $\Phi$ (%)[b] | | $\alpha_{AIE}$[c] | E (kcal/mol) [d] |
|---------|------|------|------|------|------|------|------|------|------|
| | | Soln. | Solid | Soln. | Solid | Soln. | Solid | | |
| TPO-I | 317 | 447 | 559 | 0.83 | 48740 | 0.77 | 16.00 (35.00[e]) | 20.78 | −71.43, −70.19 |
| TPO-Br | 317 | 444 | 434 549 | 0.40 | 2.52 [f] 706420 [g] | 1.08 | 17.85 (36.56[e]) | 16.53 | −85.17, −84.37 |
| TPO-Cl | 317 | 443 | 435 | 0.39 | 1.60 | 1.20 | 20.05 | 16.71 | / |
| TPO-F | 317 | 438 | 420 | 0.36 | 0.80 | 0.65 | 11.11 | 17.09 | / |
| TPO-P | 317 | 444 | 422 | 0.86 | 1.02 | 1.18 | 18.58 | 15.75 | −63.95, −71.97 |

**Table 1 Photophysical properties of AIEgens**

[a]$\tau$ = average fluorescence or phosphorescence lifetime calculated by $\tau = \Sigma A_i \tau_i^2 / \Sigma A_i \tau_i$, where $A_i$ is the pre-exponential factor for lifetime $\tau_i$
[b]$\Phi$ = fluorescence or phosphorescence quantum yield measured by using an integrating sphere
[c]$\alpha_{AIE} = \Phi_{solid} / \Phi_{soln}$
[d]E = interaction energies between anion and $\pi^+$ calculated based on their single-crystal structures by single-point calculations using the M062X/6–31+G(d,p) method
[e]Phosphorescence quantum yield of single crystal
[f]Average fluorescence lifetime at 434 nm
[g]Average phosphorescence lifetime at 549 nm

Fig. 20 and 21, Supplementary Tables 5 and 6, the higher fluorescence lifetime was observed in the solvent with larger dielectric constant due to the larger dielectric constant lead to the weaker interaction between the positive TPO core and heavy-halogen ion ($I^-$ and $Br^-$). The higher concentrations promoted the heavy atom effect for TPO-I and TPO-Br and then benefited for the intersystem crossing from the singlet to triplet states, inducing the variation of fluorescence and phosphorescence. Thus the phosphorescence was observed in their solutions with higher concentration under 77 K (Supplementary Fig. 22-24, Supplementary Tables 7-10). For their powders, the lifetime of phosphorescence and fluorescence can also be tuned by heating as exhibited in Supplementary Fig. 25-27. With increasing the temperature from 298 to 498 K, the fluorescence lifetime changed slightly while the phosphorescence lifetime decreased dramatically (Supplementary Tables 11-14). As a consequence, the fluorescence and phosphorescence properties of TPO-I and TPO-Br could be flexibly tuned.

Notably, the crystalline TPO-Br with both fluorescence at 434 nm and RTP at 549 nm is found to respond sensitively to the mechanical grinding. Before grinding, the RTP predominated in the microcrystals or powder of TPO-Br (Supplementary Fig. 28). During grinding, the phosphorescent peak at 549 nm decreased gradually with increasing the grinding time (Supplementary Fig. 29). The above changes in the emission of TPO-Br during the grinding process may be ascribed to the transition from the crystalline to the amorphous form in the solid state, supported by the XRD diffraction. As shown in Supplementary Fig. 30, the diffraction peaks in the XRD patterns suggest the degree of crystallization of microcrystals of TPO-Br decreased in intensity upon grinding. Actually, grinding caused the destruction of rigid surrounding of the crystal state for RTP of TPO-Br, resulting in the decrease of RTP efficiency[25]. Interestingly, the ratio of intensity between phosphorescence and fluorescence decreased remarkably after grinding. Thus, it may be concluded that the intensity ratio of phosphorescence and fluorescence can be easily tuned by the degree of crystallization. For TPO-I, only phosphorescence was observed before and after grinding although the intensity of XRD peaks became obviously weak (Supplementary Fig. 31 and 32), implying that the fluorescence of TPO-I was seriously quenched due to strong heavy atom effect of iodide.

**Single-crystal X-ray diffraction analysis and theoretical calculation.** To further understand the RTP phenomenon of organic salts TPO-I and TPO-Br, the single crystal analysis was conducted (Fig. 2 and Supplementary Table 15). The torsion angles between the peripheral phenyl rings and oxazole core were 26.58° ($\theta_2$), 59.98° ($\theta_3$), 81.88° ($\theta_4$), and 15.85° ($\theta_5$) for TPO-I (Fig. 2a), indicating the twisted molecular conformation in TPO-I. Notably, two iodides were located on both sides of the positively charged

oxazole core with distances of 3.980 Å and 4.071 Å (Fig. 2c). In consideration of the large radius of iodide (~2.2 Å), the strong anion–$\pi^+$ interactions between iodide and oxazole core indeed existed in the single crystal of TPO-I, similar to TPO-P in the previous work[52]. The interaction energies between iodide and oxazole core were calculated to be −71.43 and −70.19 kcal $mol^{-1}$ under the method of M062X/6-31 + G(d,p) (Table 1). For TPO-Br, the single crystal structure also beared the twisted molecular conformation with the corresponding torsion angles of 41.79°, 64.99°, 51.86°, and 37.76°. The anion–$\pi^+$ interactions were also formed in TPO-Br with the shorter distances of 3.494 Å and 3.601 Å and the larger interaction energies of −85.17 and −84.37 kcal $mol^{-1}$ (Fig. 2b, d). According to our previous strategy[52], the anion–$\pi^+$ interactions, ionization and twisted molecular conformation could efficiently block intermolecular π–π stacking to suppress aggregation-caused quenching (ACQ). Meanwhile, the short contact interactions (such as anion–$\pi^+$, C–H...π and H-bonding) suppressed the rotation of phenyl rings, inhibited the non-radiative relaxation and induced the strong aggregated emission (Supplementary Fig. 33). TPO derivatives thus exhibited strong AIE feature. Together with the EHE on the core chromophore through anion–$\pi^+$ interactions, the strong RTP for TPO-I and TPO-Br was induced in the crystalline state. In comparsion to TPO-P with fluorescence, the efficient RTP of TPO-I and TPO-Br is attributed to the effective EHE induced by heavy-atom-participated anion–$\pi^+$ interactions.

Theoretical calculations were further carried out to investigate the effect of heavy-atom-participated anion–$\pi^+$ interactions on RTP under ONIOM/B3LYP-6-311G(d)* method based on the QM/MM model (Supplementary Fig. 34). As indicated by the calculation results in Fig. 3 and Supplementary Fig. 35, the ISC process for TPO-P was hard to conduct because of the weak SOC without the aid of heavy-atom effect and wide S-T energy gap (Fig. 3a) because there are tpyical localized π → π* transitions in the involved excited singlet and triplet states (Fig. 3c). The introduction of bromine and iodine enhances SOC in TPO-Br and TPO-I owing to the heavy atom effect. Meanwhile, remarkable charge transfer character from the bromine/iodine to π-conjugated TPO core was also observed[51]. This feature not only reduces the excitation energy of $S_1/T_1$ and the energy gap between them, but also induces more excited triplet states lower than $S_1$ in energy which generates more conversion paths from excited single to triplet states.

Further, we analyzed the characters of the excited states using the nature transition orbitals (NTOs) for TPO-Br (Fig. 3d) and TPO-I (Supplementary Fig. 35). As suggested in Fig. 3d, the transitions from hole to electron NTOs for TPO-Br are dominant with proportions larger than 90% for the states. The hole NTOs extend to the two parts, including the bromine atom with n electron pairs and TPO core with π-conjugated electron, while the electron NTOs

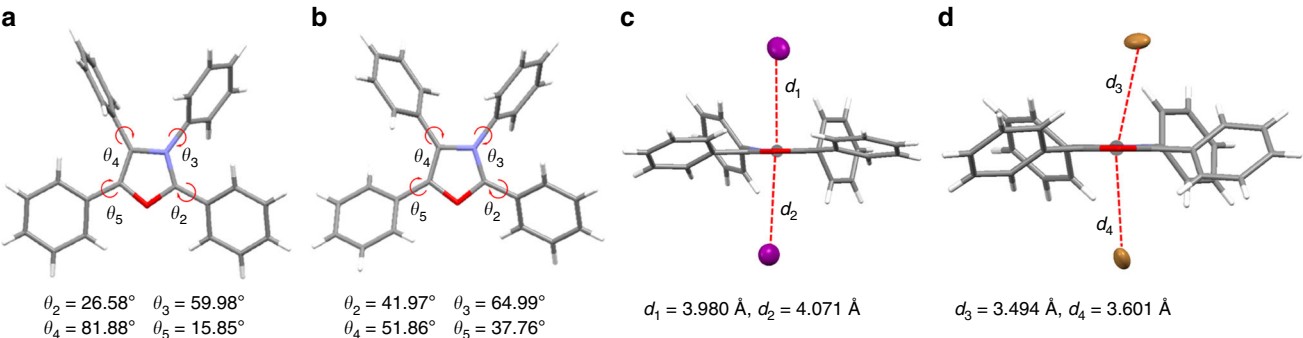

**a**
$\theta_2 = 26.58°$  $\theta_3 = 59.98°$
$\theta_4 = 81.88°$  $\theta_5 = 15.85°$

**b**
$\theta_2 = 41.97°$  $\theta_3 = 64.99°$
$\theta_4 = 51.86°$  $\theta_5 = 37.76°$

**c**
$d_1 = 3.980$ Å, $d_2 = 4.071$ Å

**d**
$d_3 = 3.494$ Å, $d_4 = 3.601$ Å

**Fig. 2** Single-crystal X-ray diffraction analysis. Torsion angles and anion–$\pi^+$ interactions with distances for **a, c** TPO-I and **b, d** TPO-Br, respectively

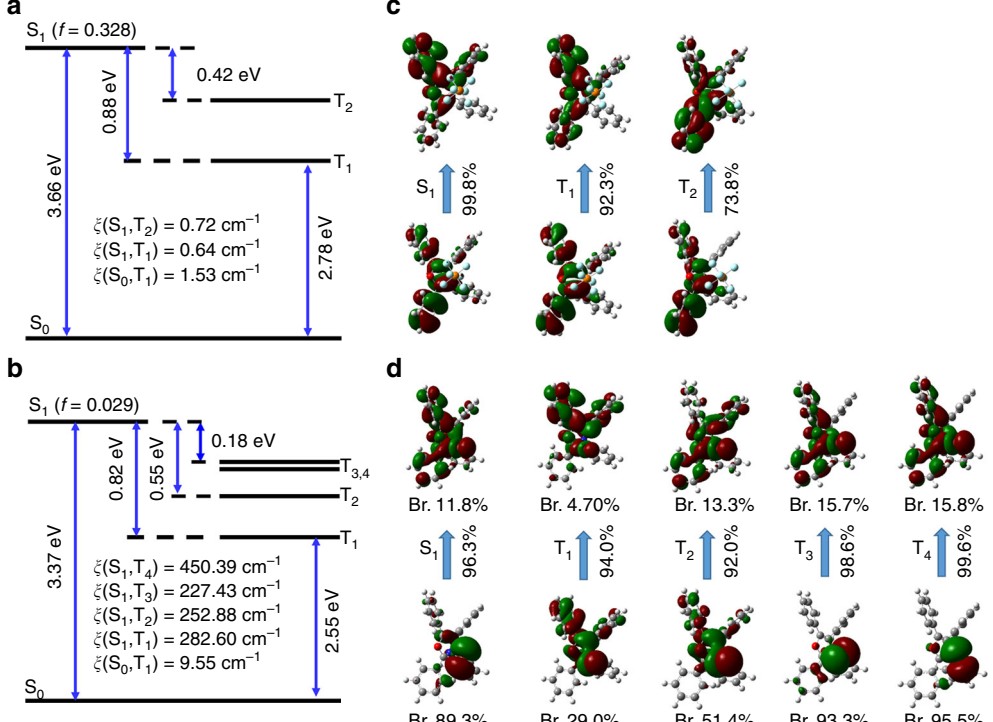

**Fig. 3** Theoretical calculation for fluorescence and RTP behaviors of organic salts. The calculated energy level diagram, spin–orbit couplings ($\xi$) between singlet and triplet states, and the oscillator strengths ($f$) of the $S_1$ state of **a** TPO-P and **b** TPO-Br in crystal based on the optimized ground-state geometries using ONIOM method. The natural transition orbitals (NTOs) (hole ones at the bottom and electron ones on the top) and the corresponding proportions for **c** TPO-P and **d** TPO-Br, as well as the bromine components in NTOs for TPO-Br

mainly concentrated on TPO core. From $T_1$, $T_2$, $T_3$ to $T_4$, the bromine component of the hole NTO increases from 29.0%, 51.4%, 93.3% to 95.5%, indicating the effect of different heavy atoms on SOC, the degree of n-$\pi^*$/$\pi$-$\pi^*$ mixture and charge transfer characters in the four triplet states. This results in the effective ISC from $S_1$ to $T_3$ and $T_4$ with large SOC and small energy gap as shown in Fig. 3b. In addition, compared to TPO-P, the oscillator strength ($f$) of $S_1$ state is sharply reduced from 0.328 to 0.029 owing to the participation of n orbital and charge transfer in TPO-Br, which weakens the fluorescence of TPO-Br. Therefore, TPO-Br shows both fluorescence and phosphorescence.

For TPO-I, the contributions of iodine to the excited states in TPO-I are far larger than those of bromine in TPO-Br, such as more outstanding heavy atom effect, which lead to much stronger phosphorescence (details in Supplementary Information). Therefore, RTP could be induced by the EHE through anion–$\pi^+$ interactions and the intensity and lifetime of RTP could be tuned by different heavy atoms to some extent.

**Construction of OSMWLEs.** The conventional RTP materials were usually consisted of a covalent structure, while the present system was composed of the fixed TPO chromophore and easily exchanged counterions with anion–$\pi^+$ interactions. Due to this unique structure, their blue fluorescence and yellow RTP can be simply tuned by these counterions to construct OSMWLEs. Above mentioned that TPO-Br shows dual emission of blue fluorescence and yellow RTP whose proportion can be tuned by the degree of crystallization[25,64], such characteristic makes TPO-Br suitable for constructing OSMWLEs. The films of TPO-Br were prepared through controlling the degree of crystallization by solution processing under different conditions of concentration and amount (Supplementary Fig. 36). Under optimum condition, the white light emission was successfully achieved with two

complementary emission at blue and yellow (Fig. 4a). The Commission Internationale de L'Eclairage (CIE) chromaticity coordinate of TPO-Br film was calculated to be (0.32, 0.33) (Fig. 4b), which is quite close to (0.33, 0.33) of pure white color defined by CIE in 1931[65]. XRD analysis for the white light emission of the TPO-Br film showed the weaker XRD diffractogram compared to its powder (Supplementary Fig. 37), suggesting the coexistence of amorphous and crystalline states with the appropriate ratio for white light emission. Additionally, the stabilities of OSMWLEs based on TPO-Br film were also investigated by continuous monitoring of the PL spectra at different temperatures (Supplementary Fig. 38 and 39). As a result, the stable PL spectra and CIE coordinate evidently demonstrated the good stability of the white light emission of TPO-Br film at room temperature, even at −20 and 40 °C.

Unlike TPO-Br, the film of TPO-I has only been observed the yellow emission like its powder (Supplementary Fig. 40). With incorporation of blue-emissive TPO-Cl into the TPO-I film, the white light emission was also envisioned to be realized through facile manipulation of the different ratios of I⁻ and Cl⁻. As shown in Fig. 4c, the emission intensity at 559 nm for TPO-I decreased gradually while it enhanced at 435 nm for TPO-Cl with increase of TPO-Cl. Under the molar ratio of 1:2, the film composed of TPO-I and TPO-Cl showed pure white light with the CIE coordinate of (0.31, 0.33) (Fig. 4d). Similar to TPO-Cl, TPO-P with blue emission could also be used to realize the construction of white light emitter. The emission of the film consisted of TPO-I and TPO-P gradually changed from yellow to blue with increasing TPO-P (Fig. 4e, f). When the molar ratio of TPO-I and TPO-P was 1:1, the film emitted white light with the CIE value of (0.30, 0.31). Therefore, the OSMWLEs obtained by these organic salts are different from conventional single-molecule and multi-components white light emitters, because

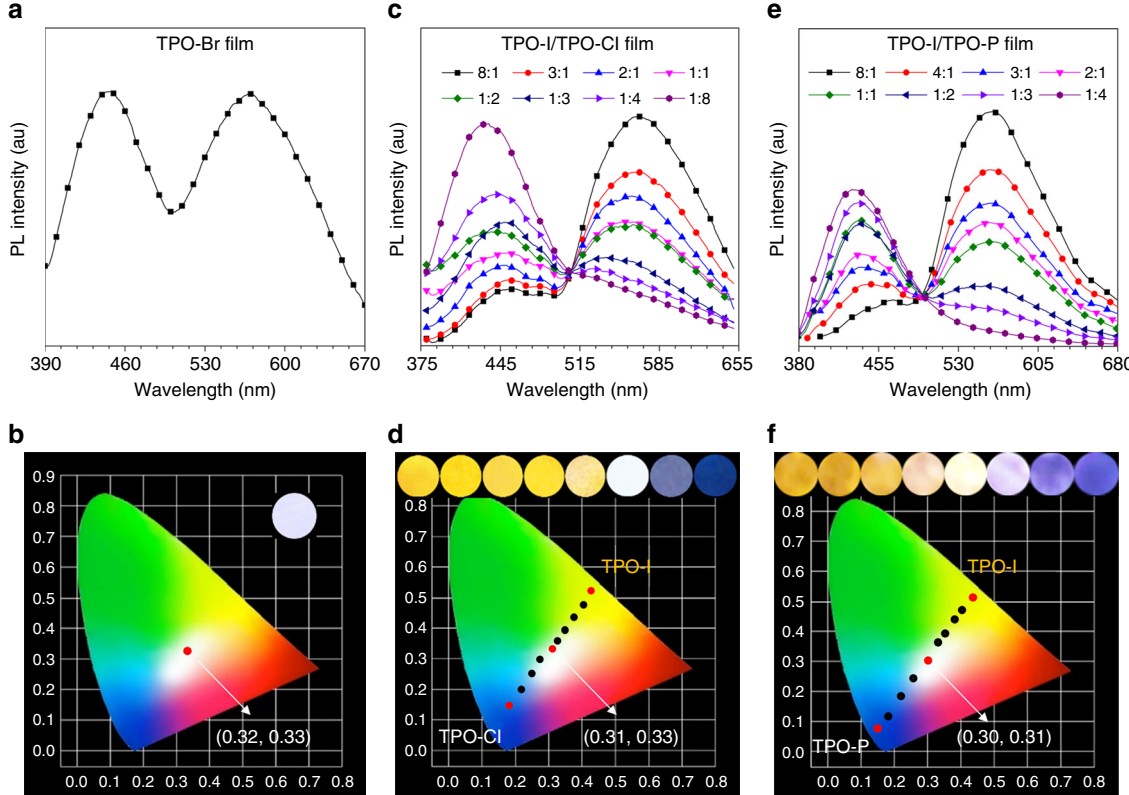

**Fig. 4** OSMWLEs constructed by organic salts. **a** TPO-Br only, **c** TPO-I/TPO-Cl and **e** TPO-I/TPO-P with different counterions molar ratios. Commission Internationale de L'Eclairage (CIE) 1931 coordinates of prompt emission of films: **b** TPO-Br only, **d** TPO-I/TPO-Cl and **f** TPO-I/TPO-P with different molar ratios. Insets show luminescent photographs of these thin films taken under 365 nm UV irradiation

the two complementary emission of fluorescence and RTP are acquired by manipulation of counterions.

**White light display and 3D printing**. For the real application, polystyrene (PS) with good film-forming ability was used as a media to encapsulate these OSMWLEs. Firstly, the PL spectrum of the PS film with TPO-Br exhibited two emission at 446 and 560 nm (Fig. 5a) and the CIE coordinate was calculated to be (0.32, 0.33) (Fig. 5b), displaying the excellent white light emission. That was also observed in the PS films containing the combination of TPO-I and TPO-Cl at molar ratio of 1:2 or TPO-I and TPO-P at molar ratio of 1:3, with the corresponding CIE coordinates of (0.31, 0.33) or (0.31, 0.34) (Fig. 5c, f). Then, we coated the TPO-Br-loaded PS film on UV-light-emitting diode (UV-LED) lamps. Upon the excitation of 365 nm from this UV-LED lamp, the white light emission can be successfully realized. As shown in Fig. 5g, when the LEDs were turned on, the LED lamp coated by TPO-Br-loaded PS film emitted white light strongly, while the other with PS film only emitted the original blue light. Because of the good film-forming ability of PS, large-area thin PS films with loading of TPO-Br were also fabricated for white light display, demonstrated by the white light Chinese character of "Tang" under 365 nm UV irradiation with the assistance of the mask (Fig. 5h).

Moreover, the thin film of TPO-Br in polyethylene glycol (PEG, average Mn 20 K) also exhibited white light emission with CIE coordinate of (0.31, 0.33) (Fig. 6a, b). As PEG can be utilized as a good 3D printing material, TPO-Br is endowed with the potential to serve as an additive of 3D printing materials for fabrication of some objectives with white light emission. To this end, taking advantages of low molecular weight PEG diacrylate

(Mn 250) with or without loading of TPO-Br and the initiator of diphenyl(2,4,6-trimethylbenzoyl)phosphine oxide, the lampshades were successfully printed by 3D printer under UV-light irradiation. Noted that the process of 3D printing and the quality of lampshades were completely unaffected by additive of TPO-Br. The lampshades are totally transparent under daylight (Fig. 6c). Without TPO-Br the lampshade exhibited weak blue emission under 365 nm UV lamp (Left in Fig. 6d) and the strong blue light was observed when the lampshade is on the turn-on UV-LED lamp (Left in Fig. 6e). To be delighted, the lampshade with TPO-Br exhibited white light both under 365 nm UV lamp and on the turn-on UV-LED lamp (Right in Fig. 6d, e), greatly demonstrating the promising potential of TPO-Br in 3D printing area.

## Discussion

EHE has been used to design RTP luminogens due to the efficient increase of SOC constant between the singlet and triplet states under non-covalent interactions, which boosts the ISC process for efficient phosphorescence. Development of effective approaches for exerting EHE on chromophores to construct RTP luminogens will be of great scientific significance and practical value. Herein, we proposed a unique strategy of heavy-atom-participated anion–$\pi^+$ interactions, which was utlized to construct RTP-active organic salts based on TPO derivatives. For instance, TPO-I and TPO-Br were induced to emit highly efficient RTP in the solid state upon the strong heavy atom effect of $I^-$ and $Br^-$ associating with the interactions of $I^- - \pi^+$ and $Br^- - \pi^+$. TPO-Br displayed dual emission of blue fluorescence and yellow RTP and their ratio of intensity can be tuned by the degree of crystallization. Only fluorescence was observed for TPO-Cl, TPO-F, and TPO-P with counterions of $Cl^-$, $F^-$, and $PF_6^-$ in the solid state without

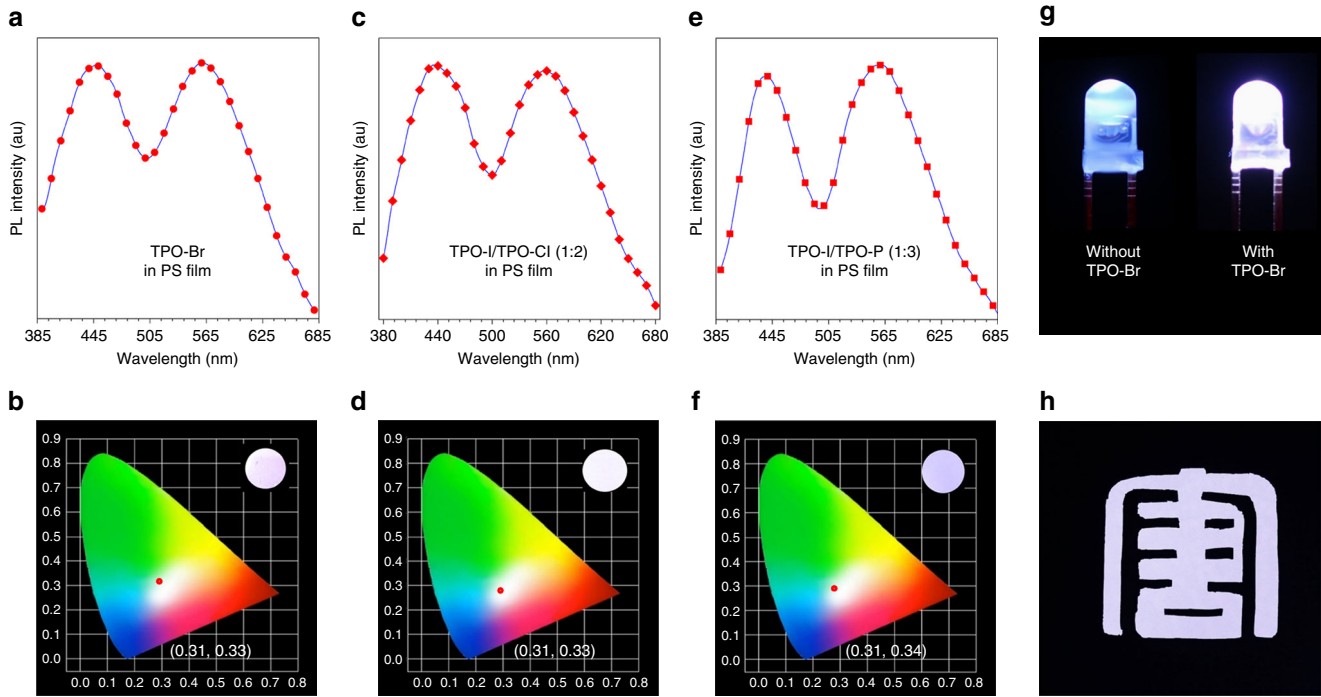

**Fig. 5** White light emitters in polystyrene (PS) films based on organic salts. PL spectra and CIE 1931 coordinates of prompt emission of **a**, **b** TPO-Br, **c**, **d** TPO-I/TPO-Cl (molar ratio = 1:2), and **e**, **f** TPO-I/TPO-P (molar ratio = 1:3) in polystyrene (PS) (2%, m/m). Insets show luminescent photographs of these PS films taken under 365 nm UV irradiation. **g** Luminescent photographs of UV-LED lamps (Emission wavelength: 360–370 nm) coated with PS films without (left) and with (right) TPO-Br. **h** Luminescent pattern of a Chinese character "Tang" based on the PS film (3 cm × 3 cm) containing TPO-Br (2%, m/m) taken under 365 nm UV irradiation

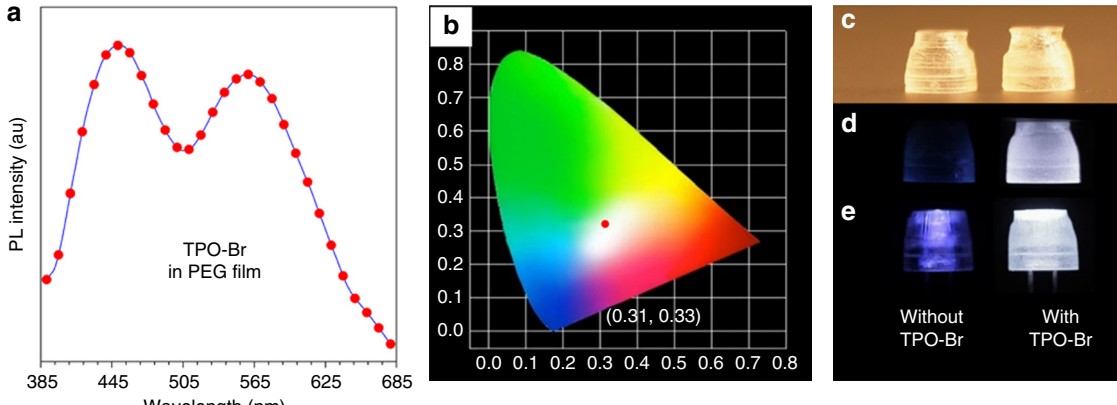

**Fig. 6** White light emitters in 3D printing. **a** PL spectra and **b** CIE 1931 coordinates of prompt emission of the PEG film containing TPO-Br (2%, m/m). Photographs of 3D printed lampshades without (left) and with (right) TPO-Br taken under **c** daylight, **d** 365 nm UV light irradiation, and **e** UV-LED lamps. The size of lampshades is 8.2 × 8.2 × 9.2 mm

obvious EHE. Theoretical calculations further verified that the RTP of TPO-I and TPO-Br was induced by the introduction of EHE via anion–π$^+$ interactions. Thus, heavy-atom-participated anion–π$^+$ interactions play a crucial role in the emergence of RTP for TPO-I and TPO-Br. It is worth to mention that the RTP properties of TPO-Br with the longer lifetime makes it well suitable for cell imaging and in vivo phosphorescent imaging with extremely low autofluorescence compared to fluorescent probes (Supplementary Figs. 41 and 42).

Owing to the excellent emission behaviors, the OSMWLEs composed of fluorescence and RTP were further established based on these organic salts. The film of TPO-Br exhibited white light

emission by simply tuning the degree of crystallization, and the film of TPO-I/TPO-Cl or TPO-I/TPO-P bearing one positively charged oxazole and two counterions also emitted white light by controlling the ratios of these two counterions. Considering the flexible operation and good machinability, polymer matrices, such as PS and PEG, were exploited to encapsulate TPO-based organic salts to construct white light emitters for the real application such as 3D printing of white light lampshades. Consequently, the facile strategy of heavy-atom-participated anion–π$^+$ interactions to construct RTP-active organic salts would greatly extend the design rationales for RTP luminogens, which will light up the enthusiasm of scientists to develop more advanced RTP

materials with multifunctionality for various high-tech applications.

## Methods

**Materials**. Chemicals were purchased from Energy-Chemical, Sigma-Aldrich, J&K and used without further purification. Solvents and other common reagents were obtained from Sigma-Aldrich. Solvents were dried and distilled out before being used for the synthesis. ¹H NMR and ¹³C NMR spectra were measured on a Bruker ARX 400 MHz spectrometer. High-resolution mass spectrometry (HRMS) were recorded on a GCT Premier CAB 048 mass spectrometer operating in MALDI-TOF mode. The starting materials and TPO derivatives were synthesized following the procedures described in the literature. Single-crystal X-ray diffraction measurements were conducted on a Bruker-Nonius Smart Apex CCD diffractometer with graphite monochromated Mo Kα radiation. Powder and film X-ray diffraction was performed using a Philips PW 1830 X-ray Diffractometer. The detail experimental procedure and synthetic methods are described in Supplementary information.

**Steady-state spectral measurements**. The steady-state absorption measurements were recorded on a Rarian 50 Conc UV–vis spectrophotometer. Photoluminescence (PL) spectra and absolute fluorescence quantum yields were measured on Fluorolog®-3 spectrofluorometer with an integrating sphere. The delayed PL spectra were measured on a PerkinElmer LS 55 spectrophotometer.

**Time-resolved luminescence measurements**. The fluorescence lifetime was measured using an Edinburgh FLSP920 fluorescence spectrophotometer equipped with a xenon laser arc lamp (Xe900), a microsecond flash lamp (uF900), and a picosecond pulsed diode laser (EPL-375), and a closed-cycle cryostate (CS202*I-DMX-1SS, Advanced Research Systems). The picosecond pulsed diode laser (EPL-375) was used as excitation source for the measurement of fluorescence lifetimes. For phosphorescence lifetimes, the excitation wavelength is 350 nm.

**Theoretical calculation**. The computational models were built from the crystal structure shown in Supplementary Figure 34. The quantum mechanics/molecular mechanics (QM/MM) theory with two-layer ONIOM method was implemented to deal with the electronic structures in crystal, where the central molecule is chosen as the active QM part and set as the high layer, while the surrounding ones are chosen as the MM part and defined as the low layer. The universal force field was used for the MM part, and the molecules of MM part were frozen during the QM/MM geometry optimizations. On the basis of the optimized geometry of the ground state ($S_0$) at B3LYP/6–311 G(d) level, the excitation energies and NTOs were calculated by using TD-DFT for electronic excited singlet and triplet states. The above results are calculated by Gaussian 09 package. At the same level, the SOCs between singlet and triplet states are given by Beijing Density Function program (details in Supplementary Information).

Based on their single crystal structures, the anion–$\pi^+$ interaction energies were calculated by single-point calculations using M062X/6–31 + G(d,p) method according to the equation $E = E_{complex} - E_{anion} - E_{cation}$, where $E =$ anion–$\pi^+$ interaction energy, $E_{complex} =$ the energy of TPO-I or TPO-Br, $E_{anion} =$ the energy of anion, $E_{cation} =$ the energy of cation. All the electronic structure calculations were performed with Gaussian 09 program package.

**Data availability**. The data that support the findings of this study are available from the authors on reasonable request, see author contributions for specific data sets. The X-ray crystallographic coordinates for the structures reported in this article have been deposited at the Cambridge Crystallographic Data Centre (CCDC) under deposition numbers CCDC: 1814784 (www.ccdc.cam.ac.uk/data_request/cif) for TPO-I, and 1814785 (www.ccdc.cam.ac.uk/data_request/cif) for TPO-Br. These data can be obtained free of charge from The Cambridge Crystallographic Data Centre.

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

## Acknowledgements

This work was partially supported by the National Natural Science Foundation of China Grant (21502022, 21788102 and 21702016), the National Basic Research Program of China (973 Program, 2013CB834701 and 2013CB834702), the Innovation and Technology Commission (ITC-CNERC14SC01), the Research Grants Council of Hong Kong (16308016, 16305015, C2014-15G, N_HKUST604/14 and A-HKUST605/16), and the Science and Technology Plan of Shenzhen (JCYJ20160229205601482). The authors thank Prof. D. Ding and his students H.Q. Gao and X. Zhang for the help of conducting bioimaging. All animal studies were performed in compliance with the guidelines set by Tianjin Committee of Use and Care of Laboratory Animals and the overall project protocols were approved by the Animal Ethics Committee of Nankai University.

## Author contributions

J.W. synthesized all materials, grew the crystals, and performed all photophysical measurements. J.W. and X.G. analyzed all experimental results, prepared and wrote the manuscript. H.M., Q.P., X.Z., G.S., and Z.S. performed the theoretical calculations and analyze calculation results. X.H. collected all single crystal data. S.H.P.S. assisted 3D printing. X.G. and J.W.Y.L. revised the paper. X.G. and B.Z.T. designed and supervised this work.

## Additional information

**Competing interests:** The authors declare no competing interests.

