## [Peer Review File · Nature Communications]

Reviewers' comments:

Reviewer #1 (Remarks to the Author):

The authors report room-temperature phosphorescent luminophores with solid-state luminescent properties. Intersystem crossing seemed to be enhanced by external heavy-atom effects via the anion- π^+ interaction. Furthermore, based on dual-emissive properties composed of blue fluorescence and yellow phosphorescence, white-light luminescent materials have been accomplished. Finally, the authors presented the applicability for OLEDs. Synthesis and characterization were performed carefully. The mechanism for inducing room-temperature phosphorescence via the ionic interaction is a key phenomenon and involves high impact especially in photochemistry. Practical demonstration would stimulate many researchers in a wide variety of fields including material science. Therefore, I judged that this manuscript deserves prompt publication in this journal once the following queries are appropriately supported.

-Please add further explanation on the mechanism of solid-state emission according to the X-ray data. Similarly to conventional AIE molecules, the twisting phenyl groups as mentioned in page 6, line 5 would play a critical role in suppressing ACQ.

-Please check again the significant digits in quantum yields.

-In page 9, line 12, please add information on durability of luminescent properties against temperature changes. If the device works much longer time, did you observe any luminochromism?

-I am interested luminescent chromism in persistence (Chem. Eur. J. 2014, 20, 8320–8324). If you have any information, please mention this issue in the main text.

Reviewer #2 (Remarks to the Author):

The authors report an interesting RTP based on anion- π^+ heavy atom effect. The organic salts of TPOs with iodine and bromine showed RTP while analogous salts with fluoride, chloride, and PF₆⁻ produce only fluorescence. Preliminary XRD and DFT calculation were performed to build understanding on the phenomena. White emission was demonstrated by TPO-br since it produces both fluorescence and phosphorescence as well as by mixing of the phosphorescent TPO-I and another fluorescent salt.

The unique thing of this report is the realization of phosphorescence by anion- π^+ heavy atom effect. However, overall the presented data and analysis are too premature to warrant publication in Nature Communications.

First of all, the role of bromine and iodine for RTP is not thoroughly studied. They claimed that the heavy halogen reduces the ST energy gap and enhances SOC. The latter is generally accepted and reasonable. However, what is the basis for the former? This reviewer cannot find relevant arguments from the cited references 25, 58, and 59. Moreover, the character of the higher T states are not analyzed. It looks that the character of T3 and T4 of TPO-Br is important since the ST gap between S1 and T1 is too large and they have the same n- π^* character, which makes the intersystem crossing a forbidden transition.

Why does TPO-I show only phosphorescence while TPO-Br produces both fluorescence and phosphorescence? A more efficient heavy atom effect of iodine promotes more efficient intersystem crossing from S1 to a higher T state? More thorough and systematic computation is necessary to build better understanding on the finding phenomena.

The authors mentioned that by tuning the aggregation modes of TPO-Br they produced white emission. This reviewer believe that this statement is misleading since they did not analyze and consequently do not know the nature of the so called "aggregation modes". They simply changed the amount of TPO-Br and concentration to make TPO-Br films and carried out simple power XRD to draw a conclusion that the film is a mixture of crystalline and amorphous TPO-Br. So, what does alter the ratio of the fluorescence and phosphorescence emission intensity is vague.

Coating of the salts on a backlight bulb to produce while emission is not attractive at all. Based on this demonstration, no one can claim that the materials are promising ones for while light emitting OLED since EL is very different from PL.

AIE is a unique phenomenon originated from the restricted molecular motion in the solid state. However, phosphorescence enhancement in the crystalline state of metal-free organic phosphors is a common phenomenon since their emission life time is so long collisional quenching is critically detrimental to RTP. Therefore, the emphasis on CIEE is meaningless.

Exerting the heavy-atom effect on luminogens through non-covalent interactions is not rare since large T1 to S1 transition has been commonly demonstrated by heavy-atom containing solvents.

Phosphorescence is very sensitive to oxygen. Oxygen sensitivity study is also recommended to understand the phenomena better.

Overall, finding RTP from the presented anion- π^+ heavy atom effect is interesting. However, the presentation of the finding is premature.

Reviewer #3 (Remarks to the Author):

Wang et al. described a class of organic salts the photoluminescence emission of which can be tuned from ns-long fluorescence to ms-long phosphorescence via the choice of halide counter ions. The photoluminescence properties of these organic salts were thoroughly investigated and the proposed mechanism backed up by calculations. Furthermore, the authors showcased an important application in single-molecule white-light emission with one of the molecules. I think this is an interesting and quality work, which is likely to attract a broad audience of chemists, material scientists, and engineers, and should merit publication in Nature Communications after appropriate revisions. I have several suggestions to make the manuscript more publishable.

1) The authors claim that "To our best knowledge, the strategy of heavy-atom-participated anion- π^+ interactions to design RTP-active organic salts has not been reported previously.", however, a literature precedent was published in J. Phys. Chem. A (2016, 120, 29, 5791-5797) on this particular strategy with similar interpretations and should be cited and discussed. Nonetheless, the current manuscript is a significant advance to the previous report in terms of the more detailed photophysical properties, theoretical and application scopes.

2) The absorption spectra in Fig. S13 show that the salts with bromide and iodide counterions are dramatically different vs. the other counterions. The authors explained such phenomenon as decreased energy gap due to cation-anion interactions (which do not show up from NMR spectra). This is surprising given that ethanol is strongly solvating. Absorption in other solvents should be checked to make sure that this shift is not due to trivial effects such as the presence of triiodine or tribromine ions.

3) In the discussion part, the authors state that "So far, it has been rarely reported to exert the heavy-atom effect on luminogens through non-covalent interactions" is inaccurate, since all external heavy-atom effects occur through non-covalent interactions (see Kasha's first JPC paper on external-heavy atom effect published in 1953 and many other in 1960s).

- 4) Regarding the spectroscopic difference between the crystal and film, the authors ascribed it to spectra from different modes of aggregation. This statement requires experimental evidence. For example, the relationship between powder XRD and photoluminescence.
- 5) The authors should list separate lifetime components and weights before adding up for average values in case some important information is missed out.

Responses to the Comments and Suggestions of Reviewer 1

Reviewer 1:

Recommendation: Minor Revision

Comments: The authors report room-temperature phosphorescent luminophores with solid-state luminescent properties. Intersystem crossing seemed to be enhanced by external heavy-atom effects via the anion- π^+ interaction. Furthermore, based on dual-emissive properties composed of blue fluorescence and yellow phosphorescence, white-light luminescent materials have been accomplished. Finally, the authors presented the applicability for OLEDs. Synthesis and characterization were performed carefully. The mechanism for inducing room-temperature phosphorescence via the ionic interaction is a key phenomenon and involves high impact especially in photochemistry. Practical demonstration would stimulate many researchers in a wide variety of fields including material science. Therefore, I judged that this manuscript deserves prompt publication in this journal once the following queries are appropriately supported.

Dear Reviewer 1:

We would like to thank the reviewer for his/her recognition of this work and the nice advices he/she made, and we revised the manuscript accordingly. Below are our point-to-point responses to the reviewer's comments.

Comment 1. *Please add further explanation on the mechanism of solid-state emission according to the X-ray data. Similarly to conventional AIE molecules, the twisting phenyl groups as mentioned in page 6, line 5 would play a critical role in suppressing ACQ.*

Our Reply: Thank you very much for your professional and kind suggestions. According to your suggestion, we have added further explanation and discussion on the mechanism of solid-state emission based on single-crystal X-ray analysis in the revised manuscript with red color highlighted as follows:

On page 8 in the revised manuscript:

“According to our previous strategy⁵², the anion- π^+ interactions, ionization and twisted molecular conformation could efficiently block intermolecular π - π stacking to suppress aggregation-caused quenching (ACQ). Meanwhile, the short contact interactions (such as anion- π^+ , C-H... π and H-

bonding) suppressed the rotation of phenyl rings, inhibited the non-radiative relaxation and induced the strong aggregated emission (Supplementary Fig. 32). TPO derivatives thus exhibited strong AIE feature.”

On page S23 in the revised Supplementary Information:

Supplementary Fig. 32 a and c Short contacts, hydrogen bonding, anion- π^+ interactions and **b** and **d** intermolecular stacking of TPO-I (top) and TPO-Br (bottom), respectively.

Comment 2. Please check again the significant digits in quantum yields.

Our Reply: We would like to thank you very much for your kind suggestions. As suggested by the reviewer, we checked the quantum yields of all AIEgens again. The quantum yields of the microcrystalline of TPO-I and TPO-Br are nearly two times as large as those of their powders, suggesting the property of crystallization-induced phosphorescence enhancement.

Comment 3. In page 9, line 12, please add information on durability of luminescent properties against temperature changes. If the device works much longer time, did you observe any luminochromism?

Our Reply: Thanks very much for your professional and kind suggestions. As suggested by the reviewer, we measured the stabilities of single-molecule white light emitter based on TPO-Br film at different temperature. The results disclose that the PL spectra and CIE coordinate of TPO-Br films are almost unchanged during 20 days at -20 and 40 °C. However, at 70 °C, the PL spectra of TPO-Br films display a slight increase in the ratio of phosphorescent/fluorescent intensity during 20 days. The CIE coordinate changed from (0.30, 0.32) to (0.32, 0.38), indicating the enhancement of phosphorescent emission. We have added further discussion in the revised manuscript with red color highlighted as follows:

On pages 10-11 in the revised manuscript:

“Additionally, the stabilities of OSMWLEs based on TPO-Br film were also investigated by continuous monitoring of the PL spectra at different temperature (Supplementary Fig. 36 and 37). As a result, the stable PL spectra and CIE coordinate evidently demonstrated the good stability of the white light emission of TPO-Br film at room temperature, even at -20 °C and 40 °C.”

On pages S27-S28 in the revised Supplementary Information:

Supplementary Fig. 37 PL spectra (**a**, **d** and **g**), CIE 1931 coordinates (**b**, **e** and **h**) and CIE value (**c**, **f** and **i**) of films of TPO-Br were monitored at (**a**, **b** and **e**) -20 °C, (**d**, **e** and **f**) 40 °C and (**g**, **h** and **i**) 70 °C during 20 days.

The stable PL spectra and CIE coordinate evidently demonstrated the good stability of the white light emission of TPO-Br film at room temperature, even at -20 °C and 40 °C. With increasing the temperature to 70 °C, the PL spectra of TPO-Br films displayed a slight increase of the ratio between phosphorescent and fluorescent intensity with the CIE coordinate changed from (0.30, 0.32) to (0.32, 0.38) during 20 days, indicating the enhancement of yellow phosphorescent emission. That is easily understood by the slow crystallization process under 70 °C.

Comment 4. *I am interested luminescent chromism in persistence (Chem. Eur. J. 2014, 20, 8320–8324). If you have any information, please mention this issue in the main text.*

Our Reply: We would like to thank you very much for your careful suggestions. As suggested by the reviewer, we carried out repeated switching of TPO-Br between amorphous and crystalline states by fuming-grinding cycle, however, the PL spectra of the ground TPO-Br powder are almost unchanged after fuming with the different solvents for 2 h. The results are consistent with the good stabilities of white light emission of TPO-Br film. The reference was cited and related discussion was further added in the revised manuscript with red color highlighted as follows:

PL spectra of ground TPO-Br powder after fuming with the different solvents for 2 h.

On page 4 in the revised manuscript:

“In addition, the microcrystalline of TPO-I and TPO-Br showed stronger emission with the quantum yields of 35.00% and 36.56% which are nearly two times as large as those of their

powders. Similar emission enhancement in the crystalline state were also reported by others⁶⁰⁻⁶².”

On page 18 in the revised manuscript:

“62. Yoshii, R., Hirose, A., Tanaka, K. & Chujo, Y. Boron diiminate with aggregation-induced emission and crystallization-induced emission-enhancement characteristics. *Chem. Eur. J.* **20**, 8320–8324 (2014).”

Responses to the Comments and Suggestions of Reviewer 2

Reviewer 2

Comments: The authors report an interesting RTP based on anion- π^+ heavy atom effect. The organic salts of TPOs with iodine and bromine showed RTP while analogous salts with fluoride, chloride, and PF₆⁻ produce only fluorescence. Preliminary XRD and DFT calculation were performed to build understanding on the phenomena. White emission was demonstrated by TPO-Br since it produces both fluorescence and phosphorescence as well as by mixing of the phosphorescent TPO-I and another fluorescent salt. The unique thing of this report is the realization of phosphorescence by anion- π^+ heavy atom effect. However, overall the presented data and analysis are too premature to warrant publication in Nature Communications.

Dear Reviewer 2:

We thank you very much for your precious time to review our manuscript and are gratefully for your appreciation of our work in term of “*The unique thing of this report is the realization of phosphorescence by anion- π^+ heavy atom effect.*”. We also thank your professional and valuable comments, which help us to improve the quality of the manuscript for publication in *Nature Communications*. According to your suggestion, we have carefully carried out the related experiment and systematic theoretical calculation to support our results. We revised the manuscript point by point as follows:

Comment 1. *First of all, the role of bromine and iodine for RTP is not thoroughly studied. They claimed that the heavy halogen reduces the ST energy gap and enhances SOC. The latter is generally accepted and reasonable. However, what is the basis for the former? This reviewer cannot find relevant arguments from the cited references 25, 58, and 59. Moreover, the character of the higher T states are not analyzed. It looks that the character of T3 and T4 of TPO-Br is important since the ST gap between S1 and T1 is too large and they have the same n- π^* character, which makes the intersystem crossing a forbidden transition.*

Our Reply: We would like to thank you very much for your professional and kind suggestions. As what the reviewer said, the introduction of bromine and iodine firstly enhances SOC owing to the heavy element effect, which is reproduced by our calculation results. At the same time, the calculated results indicate that there displays remarkable charge transfer character from the

bromine/iodine to π -conjugated TPO. This feature not only reduces the excitation energy of S_1/T_1 and their energy gap, but induces more excited triplet states lower than S_1 in energy which generates more conversion paths from excited singlet to triplet states. According to the reviewer's advice, we have added the character analyses of the higher T states for TPO-Br in Fig. 3 in revised manuscript and TPO-I in Supplementary Fig. 33.

The nature transition orbital (NTO) of the excited singlet and triplet states for TPO-Br are given in Fig. 3d in revised manuscript. As seen in Fig. 3, the transitions from hole to electron NTOs are dominant with proportions larger than 90% for the involved excited singlet and triplet states. The hole NTOs extend to the two parts containing bromine atom with n electron pairs and TPO core with π -conjugated electron, while the electron NTOs mainly concentrated on TPO core. From T_1 , T_2 , T_3 to T_4 , the bromine component of the hole NTO is increased from 29.0%, 51.4%, 93.3% to 95.5%, indicating different heavy element effect on SOC, the degree of n- π^* / π - π^* mixture and charge transfer characters in the four triplet states. This results in the effect intersystem crossing from S_1 to T_3 and T_4 with large SOC and small energy gap as shown in Fig. 3b. In addition, relative to TPO-P, the oscillator strength of S_1 state is sharply reduced from 0.328 to 0.029 owing to the participation of n orbital and charge transfer feature in TPO-Br, which weakens the fluorescence of TPO-Br. Therefore, TPO-Br shows both fluorescence and phosphorescence.

As seen in Supplementary Fig. 33, the contributions of iodine to the excited states in TPO-I are far larger than those of bromine in TPO-Br, and more outstanding heavy element effect and charge transfer characters are exhibited. This further promotes the efficiency of intersystem crossing from singlet to triplet and quenches the fluorescence to some extent.

On pages 8-9, the corresponding revision has been done in the revised manuscript with highlighted in red:

Fig. 3 Theoretical calculation for fluorescence and RTP behaviors of organic salts. The calculated energy level diagram, spin-orbit couplings (ξ) between singlet and triplet states, and the oscillator strengths (f) of the S_1 state of **a** TPO-P and **b** TPO-Br in crystal based on the optimized ground-state geometries using ONIOM method. The natural transition orbitals (NTOs) (hole ones at the bottom and electron ones on the top) and the corresponding proportions for **c** TPO-P and **d** TPO-Br, as well as the bromine components in NTOs for TPO-Br.

Theoretical calculations were further carried out to investigate the effect of heavy-atom-participated anion- π^+ interactions on RTP under ONIOM/B3LYP-6-311G* method. As indicated by the calculation results in Fig. 3 and Supplementary Fig. 33, the ISC process for TPO-P was hard to conduct because of the weak SOC without the aid of heavy-atom effect and wide S-T energy gap (Fig. 3a) because there are typical localized $\pi \rightarrow \pi^*$ transitions in the involved excited singlet and triplet states (Fig. 3c). The introduction of bromine and iodine firstly enhances SOC in TPO-Br and TPO-I owing to the heavy atom effect. Meanwhile, it was also revealed that there displays remarkable charge transfer character from the bromine/iodine to π -conjugated TPO core⁵¹. This feature not only reduces the excitation energy of S_1/T_1 and the energy gap between them, but also induces more excited triplet states lower than S_1 in energy which generates more conversion paths from excited single to triplet states.

Further, we analyzed the characters of the excited states using the nature transition orbitals (NTOs) for TPO-Br (Fig. 3d) and TPO-I (Supplementary Fig. 33). As suggested in Fig. 3d, the transitions from hole to electron NTOs for TPO-Br are dominant with proportions larger than 90% for the states. The hole NTOs extend to the two parts, including the bromine atom with n electron pairs and TPO core with π -conjugated electron, while the electron NTOs mainly concentrated on TPO core. From T_1 , T_2 , T_3 to T_4 , the bromine component of the hole NTO increases from 29.0%, 51.4%, 93.3% to 95.5%, indicating different heavy atom effect on SOC, the degree of n - π^* / π - π^* mixture and charge transfer characters in the four triplet states. This results in the effective ISC from S_1 to T_3 and T_4 with large SOC and small energy gap as shown in Fig. 3b. In addition, compared to TPO-P, the oscillator strength (f) of S_1 state is sharply reduced from 0.328 to 0.029 owing to the participation of n orbital and charge transfer in TPO-Br, which weakens the fluorescence of TPO-Br. Therefore, TPO-Br shows both fluorescence and phosphorescence.

For TPO-I, the contributions of iodine to the excited states in TPO-I are far larger than those of bromine in TPO-Br, such as more outstanding heavy atom effect and charge transfer characters, which lead to much stronger phosphorescence (details in Supplementary Information). Therefore, RTP could be induced by the EHE through anion- π^+ interactions and the intensity and lifetime of RTP could be tuned by different heavy atoms to some extent.

**On pages S23-S25, Supplementary Fig. 33 in the revised Supplementary Information:
Computational Details**

The computational models were built from the crystal structure shown in Supplementary Chart 1. The quantum mechanics/molecular mechanics (QM/MM) theory with two-layer ONIOM method was implemented to deal with the electronic structures in crystal, where the central molecule is chosen as the active QM part and set as the high layer, while the surrounding ones are chosen as the MM part and defined as the low layer. The universal force field (UFF) was used for the MM part, and the molecules of MM part were frozen during the QM/MM geometry optimizations. On the basis of the optimized geometry of the ground state (S_0) at B3LYP/6-311G(d) level, the excitation energies and nature transition orbitals (NTOs) were calculated by using TD-DFT for electronic excited singlet and triplet states. The above results are calculated by Gaussian 09 package.¹ At the same level, the spin-orbit couplings between singlet and triplet states are given by Beijing Density Function (BDF) program.²⁻⁴

Supplementary Chart 1. QM/MM model taking TPO-Br as an example: one central QM molecule for the higher layer and the remain MM molecules for the lower layer.

Supplementary Fig. 33 The calculated energy level diagram, spin-orbit couplings (ξ) between singlet and triplet states, and the oscillator strengths (f) of the S_1 state of **a** TPO-I in crystal based on the optimized ground-state geometries using ONIOM method. The natural transition orbitals (NTOs) (hole ones at the bottom and electron ones on the top) and the corresponding proportions for **b** TPO-I.

For TPO-I, the contributions of iodine to the excited states in TPO-I are far larger than those of bromine in TPO-Br, and more outstanding heavy atom effect and charge transfer characters are exhibited (Supplementary Fig. 33). Those have more effects on the photophysical properties of TPO-I. The complete charge transfer from iodine atom to TPO core (i) forbids the electric dipole transition from the S_1 to S_0 state, which totally quenches the fluorescence; (ii) extremely reduces the S-T energy gap, which greatly facilitates the ISC from the S_1 to triplet states; and (iii) introduces more efficient heavy-atom effect, which further promotes more efficient ISC from the S_1 to triplet states. Thus, TPO-I is observed phosphorescent only. It should be noted that unfortunately the calculated excitation energies of the excited states all are underestimated when compared with the experimental values owing to the defect of theoretical method.

Comment 2. *Why does TPO-I show only phosphorescence while TPO-Br produces both fluorescence and phosphorescence? A more efficient heavy atom effect of iodine promotes more efficient intersystem crossing from S_1 to a higher T state? More thorough and systematic computation is necessary to build better understanding on the finding phenomena.*

Our Reply: We well agree with what the reviewer said and have added the systematic computation of TPO-I in Supplementary Fig. 33. As seen in Supplementary Fig. 33, the contributions of iodine to the excited states in TPO-I are far larger than those of bromine in TPO-Br, and more outstanding heavy element effect and complete charge transfer characters are exhibited (*J. Phys. Chem. A* **2016**, *120*, 5791–5797). Those have more effects on the photophysical properties of TPO-I. Firstly, the complete charge transfer from iodine atom to TPO part (i) forbids the dipole transition in the S_1 state, which quenches the fluorescence; (ii) extremely reduces the S-T gap, which greatly facilitates the intersystem crossing from singlet to triplet; and (iii) introduces more efficient heavy atom effect, which further promotes more efficient intersystem crossing from singlet to triplet. In addition, it should be noted that unfortunately the calculated excitation energies of the excited states all are underestimated when compared with the experimental values owing to the defect of theoretical method.

Comment 3. *The authors mentioned that by tuning the aggregation modes of TPO-Br they produced white emission. This reviewer believe that this statement is misleading since they did*

not analyze and consequently do not know the nature of the so called “aggregation modes”. They simply changed the amount of TPO-Br and concentration to make TPO-Br films and carried out simple power XRD to draw a conclusion that the film is a mixture of crystalline and amorphous TPO-Br. So, what does alter the ratio of the fluorescence and phosphorescence emission intensity is vague.

Our Reply: Thank you very much for your careful suggestions. To express our meaning clearly, as you recommended, the “aggregation modes” have been replaced by “the degree of crystallization”. In order to illuminate what altered the ratio of the fluorescence and phosphorescence emission intensity, the PL spectra and XRD patterns of TPO-Br microcrystals were measured at different grinding times. With the increase of grinding time, the ratio of intensity between phosphorescence and fluorescence and the intensity of XRD peaks gradually decrease. Inspired by a similar literature (*Angew. Chem. Int. Ed.* 2015, 54, 6270–6273), the results showed that the crystals of TPO-Br were converted into the amorphous powder by mechanical stimulus. In addition, in order to be insight into the behaviors of fluorescence and phosphorescence of TPO-I and TPO-Br, we carefully studied the properties of fluorescence and phosphorescence under different conditions such as dielectric constants of solvents, concentration and temperature. All the revisions were highlighted in red color in the revised manuscript as follows:

On pages 6-7 in the revised manuscript with highlighted in red color:

“Further, we continued to investigate the photophysical properties of TPO-I and TPO-Br under different conditions such as dielectric constants of solvents, concentration and temperature. As shown in Supplementary Fig. 19 and 20, Supplementary Tables 5 and 6, the higher fluorescence lifetime was observed in the solvent with larger dielectric constant due to the larger dielectric constant lead to the weaker interaction between the positive TPO core and heavy-halogen ion (I^- and Br^-). The higher concentrations promoted the heavy atom effect for TPO-I and TPO-Br and then benefited for the intersystem crossing from the singlet to triplet states, inducing the variation of fluorescence and phosphorescence. Thus the phosphorescence was observed in their solutions with higher concentration under 77 K (Supplementary Fig. 21-23, Supplementary Tables 7-10). For their powders, the lifetime of phosphorescence and fluorescence can also be tuned by heating as exhibited in Supplementary Fig. 24-26. With

increasing the temperature from 298 K to 498 K, the fluorescence lifetime changed slightly while the phosphorescence lifetime decreased dramatically (Supplementary Tables 11-14). As a consequence, the fluorescence and phosphorescence properties of TPO-I and TPO-Br could be flexibly tuned.

Notably, the crystalline TPO-Br with both fluorescence at 434 nm and RTP at 549 nm is found to respond sensitively to the mechanical grinding. Before grinding, the RTP predominated in the microcrystals or powder of TPO-Br (Supplementary Fig. 27). During grinding, the phosphorescent peak at 549 nm decreased gradually with increasing the grinding time (Supplementary Fig. 28). The above changes in the emission of TPO-Br during the grinding process may be ascribed to the transition from the crystalline to the amorphous form in the solid state, supported by the XRD diffraction. As shown in Supplementary Fig. 29, the diffraction peaks in the XRD patterns suggest the degree of crystallization of microcrystals of TPO-Br decreased in intensity upon grinding. Actually, grinding caused the destruction of rigid surrounding of the crystal state for RTP of TPO-Br, resulting in the decrease of RTP efficiency²⁵. Interestingly, the ratio of intensity between phosphorescence and fluorescence decreased remarkably after grinding. Thus, it may be concluded that the intensity ratio of phosphorescence and fluorescence can be easily tuned by the degree of crystallization. For TPO-I, only phosphorescence was observed before and after grinding although the intensity of XRD peaks became obviously weak (Supplementary Fig. 30 and 31), implying that the fluorescence of TPO-I was seriously quenched due to strong heavy atom effect of iodide.”

On pages S13-S21 in the revised Supplementary Information:

Supplementary Fig. 19 PL spectra of **a** TPO-I and **b** TPO-Br in different solvents (THF and MeOH) with different dielectric constants at room temperature. The dielectric constants of THF and MeOH are 7.58 and 32.7, respectively. Their emission peaks are at about 450 nm.

Supplementary Fig. 20 Time-resolved PL decay of **a** TPO-I (@ 450 nm) and **b** TPO-Br (@ 450 nm) in different solvents at the room temperature ($\lambda_{\text{ex}} = 375$ nm) under air.

Supplementary Table 5. Fluorescence lifetime data of TPO-I in different solvents with different dielectric constants at room temperature ($\lambda_{\text{ex}} = 375 \text{ nm}$, $\lambda_{\text{em}} = 450 \text{ nm}$) under air.

	THF ^b	MeOH ^b
τ_1 (ns) / (%)	4.37 / 3.44	4.47 / 8.96
τ_2 (ns) / (%)	0.31 / 96.56	0.28 / 91.04
τ (μs) ^a	1.67	2.84

^a τ = average fluorescence lifetime calculated by $\tau = \Sigma A_i \tau_i^2 / \Sigma A_i \tau_i$, where A_i is the pre-exponential factor for lifetime τ_i . ^b The dielectric constants of THF and MeOH are 7.58 and 32.7, respectively.

Supplementary Table 6. Fluorescence lifetime data of TPO-Br in different solvents with different dielectric constants at room temperature ($\lambda_{\text{ex}} = 375 \text{ nm}$, $\lambda_{\text{em}} = 450 \text{ nm}$) under air.

	THF	MeOH
τ_1 (ns) / (%)	4.98 / 0.01	0.37 / 94.65
τ_2 (ns) / (%)	0.30 / 99.99	6.17 / 5.35
τ (ns) ^a	0.30	3.18

^a τ = average fluorescence lifetime calculated by $\tau = \Sigma A_i \tau_i^2 / \Sigma A_i \tau_i$, where A_i is the pre-exponential factor for lifetime τ_i . ^b The dielectric constants of THF and MeOH are 7.58 and 32.7, respectively.

Supplementary Fig. 21 PL spectra of **a** TPO-I and **b** TPO-Br with different concentrations at 300 K and 77 K.

Supplementary Fig. 22 Time-resolved PL decay of **a** TPO-I (@ 447 nm) and **b** TPO-Br (@ 444 nm) with the different concentrations in MeOH at 300 K and 77 K ($\lambda_{\text{ex}} = 375$ nm) under air.

Supplementary Table 7. Fluorescence lifetime data of TPO-I with different concentrations in MeOH at 300 K and 77 K ($\lambda_{\text{ex}} = 375$ nm, $\lambda_{\text{em}} = 447$ nm) under air.

	1×10^{-2} (mol/L)		1×10^{-3} (mol/L)		1×10^{-4} (mol/L)		1×10^{-5} (mol/L)	
	300 K	77 K	300 K	77 K	300 K	77 K	300 K	77 K
τ_1 (ns) /	0.23 /	3.91 /	3.99 /	4.76 /	0.29 /	4.70 /	4.47 /	1.71 /
(%)	96.46	18.86	1.85	48.79	98.50	45.19	8.96	21.56
τ_2 (ns) /	3.25 /	1.16 /	0.26 /	1.22 /	4.86 /	1.18 /	0.28 /	5.01 /
(%)	3.54	81.14	98.15	51.21	1.50	54.81	91.04	78.44
τ (ns) ^a	1.26	2.37	1.10	4.01	1.22	3.88	2.84	4.72

^a τ = average fluorescence lifetime calculated by $\tau = \sum A_i \tau_i^2 / \sum A_i \tau_i$, where A_i is the pre-exponential factor for lifetime τ_i .

Supplementary Table 8. Fluorescence lifetime data of TPO-Br with different concentrations in MeOH at 300 K and 77 K ($\lambda_{\text{ex}} = 375$ nm, $\lambda_{\text{em}} = 444$ nm) under air.

	1×10^{-2} (mol/L)		1×10^{-3} (mol/L)		1×10^{-4} (mol/L)		1×10^{-5} (mol/L)	
	300 K	77 K	300 K	77 K	300 K	77 K	300 K	77 K
τ_1 (ns) /	0.2 /	3.63 /	5.48 /	4.37 /	7.00 /	4.99 /	0.37 /	5.01 /
(%)	97.99	12.42	2.08	26.89	2.06	75.88	94.65	31.43
τ_2 (ns) /	2.34 /	1.15 /	0.27 /	1.16 /	0.32 /	1.44 /	6.17 /	0.66 /
(%)	2.01	87.58	97.91	73.11	97.94	24.12	5.35	68.57
τ (ns) ^a	0.61	1.92	1.84	3.02	2.43	4.69	3.18	4.03

^a τ = average fluorescence lifetime calculated by $\tau = \sum A_i \tau_i^2 / \sum A_i \tau_i$, where A_i is the pre-exponential factor for lifetime τ_i .

Supplementary Fig. 23 Time-resolved PL decay of **a** TPO-I (@ 559 nm) and **b** TPO-Br (@ 549 nm) with the different concentrations in MeOH at 77 K ($\lambda_{\text{ex}} = 350$ nm) under air.

Supplementary Table 9. Phosphorescence lifetime data of TPO-I with different concentrations in MeOH at 300 K and 77 K ($\lambda_{\text{ex}} = 350$ nm, $\lambda_{\text{em}} = 559$ nm) under air.

	1×10^{-2} (mol/L)		1×10^{-3} (mol/L)		1×10^{-4} (mol/L)		1×10^{-5} (mol/L)	
	300 K	77 K	300 K	77 K	300 K	77 K	300 K	77 K
τ_1 (μs) / (%)	ND ^a	365 / 15.42	ND	256 / 12.07	ND	100 / 2.19	ND	7.5 / 100
τ_2 (μs) / (%)	ND	125 / 43.67	ND	57 / 16.11	ND	7 / 96.52	ND	/
τ_3 (μs) / (%)	ND	35 / 40.91	ND	6 / 71.82	ND	371 / 1.29	ND	/
τ (μs) ^b	ND	222	ND	199	ND	149	ND	7.5

^a No detectable. ^b τ = average phosphorescence lifetime calculated by $\tau = \Sigma A_i \tau_i^2 / \Sigma A_i \tau_i$, where A_i is the pre-exponential factor for lifetime τ_i .

Supplementary Table 10. Phosphorescence lifetime data of TPO-Br with different concentrations in MeOH at 300 K and 77 K ($\lambda_{\text{ex}} = 350$ nm, $\lambda_{\text{em}} = 549$ nm) under air.

	1×10^{-2} (mol/L)		1×10^{-3} (mol/L)		1×10^{-4} (mol/L)		1×10^{-5} (mol/L)	
	300 K	77 K	300 K	77 K	300 K	77 K	300 K	77 K
τ_1 (μs) / (%)	ND ^a	5.6 / 100	ND	ND	ND	ND	ND	ND
τ_2 (μs) / (%)	ND	/	ND	ND	ND	ND	ND	ND
τ (μs) ^b	ND	5.6	ND	ND	ND	ND	ND	ND

^a No detectable. ^b τ = average phosphorescence lifetime calculated by $\tau = \Sigma A_i \tau_i^2 / \Sigma A_i \tau_i$, where A_i is the pre-exponential factor for lifetime τ_i .

Supplementary Fig. 24 PL spectra of TPO-I powder at different temperature.

Supplementary Fig. 25 a PL spectra and **b** phosphorescence (573 nm)/fluorescence (448 nm) intensity ratio of TPO-Br powder at different temperature.

Supplementary Fig. 26 Time-resolved PL decay of powders of TPO-I **a** @ 450 nm, **c** @ 559 nm and TPO-Br **b** @ 434 nm, **d** @ 549 nm at different temperature under air.

Supplementary Table 11. Fluorescence lifetime data of TPO-I powder at different temperature under air ($\lambda_{\text{ex}} = 375 \text{ nm}$, $\lambda_{\text{em}} = 450 \text{ nm}$).

	298 K	348 K	398 K	448 K	498 K
τ_1 (ns) / (%)	0.14 / 20.86	0.11 / 20.59	0.09 / 25.22	0.06 / 30.36	0.05 / 29.22
τ_2 (ns) / (%)	1.55 / 79.14	1.47 / 79.41	1.59 / 74.78	1.21 / 69.64	1.99 / 70.78
τ (ns) ^a	1.52	1.44	1.19	1.18	1.97

^a τ = average fluorescence lifetime calculated by $\tau = \Sigma A_i \tau_i^2 / \Sigma A_i \tau_i$, where A_i is the pre-exponential factor for lifetime τ_i .

Supplementary Table 12. Fluorescence lifetime data of TPO-Br powder at different temperature under air ($\lambda_{\text{ex}} = 375 \text{ nm}$, $\lambda_{\text{em}} = 434 \text{ nm}$).

	298 K	348 K	398 K	448 K	498 K
τ_1 (ns) / (%)	0.05 / 88.28	0.04 / 89.58	0.08 / 85.31	0.02 / 89.88	0.02 / 91.83
τ_2 (ns) / (%)	0.90 / 11.72	0.80 / 10.42	0.59 / 14.69	0.62 / 10.12	0.60 / 8.17
τ (ns) ^a	0.65	0.57	0.37	0.49	0.44

^a τ = average fluorescence lifetime calculated by $\tau = \Sigma A_i \tau_i^2 / \Sigma A_i \tau_i$, where A_i is the pre-exponential factor for lifetime τ_i .

Supplementary Table 13. Phosphorescence lifetime data of TPO-I powder at different temperature under air ($\lambda_{\text{ex}} = 350 \text{ nm}$, $\lambda_{\text{em}} = 559 \text{ nm}$).

	298 K	348 K	398 K	448 K	498 K
τ_1 (μs) / (%)	30.27 / 36.38	27.16 / 71.82	7.02 / 14.81	/	/
τ_2 (μs) / (%)	61.03 / 63.62	58.13 / 28.18	22.10 / 85.19	/	/
τ (μs) ^a	54.23	41.30	21.31	/	/

^a τ = average phosphorescence lifetime calculated by $\tau = \Sigma A_i \tau_i^2 / \Sigma A_i \tau_i$, where A_i is the pre-exponential factor for lifetime τ_i .

Supplementary Table 14. Phosphorescence lifetime data of TPO-Br powder at different temperature under air ($\lambda_{\text{ex}} = 350 \text{ nm}$, $\lambda_{\text{em}} = 549 \text{ nm}$).

	298 K	348 K	398 K	448 K	498 K
τ_1 (μs) / (%)	269 / 35.69	95 / 47.80	31 / 38.28	5 / 47.65	3 / 50.02
τ_2 (μs) / (%)	715 / 64.31	284 / 52.20	83 / 61.72	13 / 52.35	12 / 49.98
τ (μs) ^a	637	240	73	11	10

^a τ = average phosphorescence lifetime calculated by $\tau = \Sigma A_i \tau_i^2 / \Sigma A_i \tau_i$, where A_i is the pre-exponential factor for lifetime τ_i .

Supplementary Fig. 27 PL spectra of powder and microcrystal of TPO-Br.

Supplementary Fig. 28 a PL spectra and **b** the intensity ratio between phosphorescence (549 nm) and fluorescence (434 nm) for TPO-Br microcrystals under grinding for different time.

Supplementary Fig. 29 XRD patterns of TPO-Br microcrystals under grinding for different time.

Supplementary Fig. 30 PL spectra of TPO-I powder under grinding for different time.

Supplementary Fig. 31 XRD patterns of TPO-I powder before and after grinding for 8 min.

Comment 4. *Coating of the salts on a backlight bulb to produce white emission is not attractive at all. Based on this demonstration, no one can claim that the materials are promising ones for white light emitting OLED since EL is very different from PL.*

Our Reply: Thank you very much for your suggestions. Indeed, as the reviewer said, EL is very different from PL, lots of luminescent materials are well photoluminescent but not suitable for electroluminescent application such as OLED. In this work, we have not demonstrated that these luminescent salts are promising for white light emitting OLED. We made use of a UV-LED bulb coating with these salts to simply obtain the white light emission upon the excitation of UV-LED bulb, demonstrating that these salts could be potential for decorative lighting with white light because of its advantages of simple, cheap and flexible.

Comment 5. *AIE is a unique phenomenon originated from the restricted molecular motion in the solid state. However, phosphorescence enhancement in the crystalline state of metal-free organic phosphors is a common phenomenon since their emission life time is so long collisional quenching is critically detrimental to RTP. Therefore, the emphasis on CIEE is meaningless.*

Our Reply: Thank you very much for your kind reminder. The phosphorescence enhancement in the crystalline state of metal-free organic phosphors is really a common phenomenon. In our manuscript, we also observed this phenomenon and tuned emission color of TPO-Br by grinding or film-forming. As you recommended, the discussion on CIEE has been deleted because it is a common phenomenon for phosphorescence. Moreover, in order to make this point more clearly, we have added a brief explanation in the revised manuscript with red color highlighted as follows:

On page 4 in the revised manuscript:

“In addition, the microcrystalline of TPO-I and TPO-Br showed stronger emission with the quantum yields of 35.00% and 36.56% which are nearly two times as large as those of their powders. **Similar emission enhancement in the crystalline state were also reported by others⁶⁰⁻⁶².**”

Comment 6. *Exerting the heavy-atom effect on luminogens through non-covalent interactions is not rare since large T1 to S1 transition has been commonly demonstrated by heavy-atom containing solvents.*

Our Reply: Thank you very much for your reminder. According to your suggestion, in order to more precisely illustrate our design rationale and experimental results, we have added a more reasonable presentation in the introduction part of the revised manuscript with red color highlighted as follows:

On pages 2-3 in the revised manuscript:

“Up to now, some pure organic RTP luminogens have been developed involving mainly keto^{26, 31, 40}, carbazole^{17, 33, 41, 42} and borate⁴³⁻⁴⁶ functionalities, **and their RTP properties were generally tuned by heavy atom effect because of the efficient enhancement of spin-orbit coupling (SOC) to promote the ISC process. Among them, external heavy atom effect (EHE) has been extensively studied since the early 1950s⁴⁷. Many efforts have been devoted to understand the nature of EHE because it is of essential importance to RTP in molecular design⁴⁸⁻⁵⁰. Although the EHE has routinely served as an effective strategy to promote the occurrence of phosphorescence⁵¹, it is still of special interest to explore effective approaches to exert EHE on the design of RTP luminogens.**”

“If the counterions were exchanged by heavy halide ions such as iodide (I^-) and bromide ion (Br^-), RTP-active organic salts were envisioned to be prepared based on the EHE via anion- π^+ interactions, opening a new avenue to design pure organic RTP luminogens. Moreover, the emission of these molecules would be expected to be manipulated to the desired application in OSMWLEs.”

“Due to the anion- π^+ interactions between the heavy halide ions and the positively-charged aromatic ring, the external heavy halide ions are drawn to be closed to the core chromophore to enhance the SOC to boost the ISC process effectively. Such kind of interactions was defined as heavy-atom-participated anion- π^+ interactions, which could serve as a novel strategy to design RTP luminogens.”

Comment 7. *Phosphorescence is very sensitive to oxygen. Oxygen sensitivity study is also recommended to understand the phenomena better.*

Our Reply: Thanks for your professional suggestion. As you recommended, we measured the phosphorescence lifetimes of organic salts in vacuum. The results disclose that the phosphorescence lifetimes were only slightly increased. Probably, the influence of oxygen on phosphorescent lifetime was inhibited due to strong heavy-atom-participated anion- π^+ interactions in the rigid crystalline state. All the revisions were highlighted in red color in the revised manuscript as follows:

On page 6 in the revised manuscript:

“In addition, the lifetimes of TPO-I and TPO-Br at 559 and 549 nm in vacuum were also evaluated with the average values of 52 μs and 763 μs , respectively (Supplementary Fig. 18 and Supplementary Table 4), which were slightly larger than that in air, suggesting that such phosphorescence lifetimes of TPO-I and TPO-Br in the solid state were negligibly sensitive to oxygen.”

On pages S12-S13 in the revised Supplementary Information:

Supplementary Fig. 18 Time-resolved PL decay of TPO-I (@ 559 nm) and TPO-Br (@ 549 nm) in powder in vacuum.

Supplementary Table 4. Phosphorescence lifetime data of AIEgens in powder at room temperature (298 K; $\lambda_{\text{ex}} = 350$ nm) under air and in vacuum.

	TPO-I		TPO-Br	
	under air	in vacuum	under air	in vacuum
λ_{em} (nm)	559	559	549	549
τ_1 (μs) / A_1 (%)	66.09 / 41.30	9.42 / 2.24	204.00 / 20.14	92.31 / 1.94
τ_2 (μs) / A_2 (%)	27.66 / 58.70	54.65 / 77.76	743.12 / 76.25	388.80 / 4.74
τ_3 (μs) / A_3 (%)	/	/	5.79 / 3.61	858.40 / 63.32
τ (μs) ^a	48.74	52.52	706.42	763.16

^a τ = average phosphorescence lifetime calculated by $\tau = \sum A_i \tau_i^2 / \sum A_i \tau_i$, where A_i is the pre-exponential factor for lifetime τ_i .

Responses to the Comments and Suggestions of Reviewer 3

Reviewer 3:

Recommendation: Minor Revision

Comments: Wang et al. described a class of organic salts the photoluminescence emission of which can be tuned from ns-long fluorescence to ms-long phosphorescence via the choice of halide counter ions. The photoluminescence properties of these organic salts were thoroughly investigated and the proposed mechanism backed up by calculations. Furthermore, the authors showcased an important application in single-molecule white-light emission with one of the molecules. I think this is an interesting and quality work, which is likely to attract a broad audience of chemists, material scientists, and engineers, and should merit publication in Nature Communications after appropriate revisions. I have several suggestions to make the manuscript more publishable.

Dear Reviewer 3:

We thank you for your precious time to review our manuscript and are grateful for his/her recognition of this work and the nice advice he/she made, and we revised the manuscript accordingly. Below are our point-to-point responses to the reviewer's comments.

Comment 1. *The authors claim that "To our best knowledge, the strategy of heavy-atom-participated anion- π^+ interactions to design RTP-active organic salts has not been reported previously.", however, a literature precedent was published in *J. Phys. Chem. A* (2016, 120, 29, 5791-5797) on this particular strategy with similar interpretations and should be cited and discussed. Nonetheless, the current manuscript is a significant advance to the previous report in terms of the more detailed photophysical properties, theoretical and application scopes.*

Our Reply: Thank you very much for your professional suggestions. According to your suggestion, in order to more precisely illustrate our experimental results, we have added a more reasonable presentation in the introduction part of the revised manuscript and the reference was discussed and cited.

On pages 2-3 in the revised manuscript with highlighted in red color:

“The current popular strategies to achieve pure organic RTP luminogens are to introduce heavy atoms, heteroatoms (N, O, S, P, and so on) or charge transfer state into the luminescent skeletons to facilitate the effective intersystem crossing (ISC)²⁸⁻³¹ and to modulate the aggregation behaviors to suppress the non-radiative dissipation by polymer aggregation³², crystallization³³⁻³⁶, or supramolecular assembly³⁷⁻³⁹. Up to now, some pure organic RTP luminogens have been developed involving mainly keto^{26, 31, 40}, carbazole^{17, 33, 41, 42} and borate⁴³⁻⁴⁶ functionalities, and their RTP properties were generally tuned by heavy atom effect because of the efficient enhancement of spin-orbit coupling (SOC) to promote the ISC process. Among them, external heavy atom effect (EHE) has been extensively studied since the early 1950s⁴⁷. Many efforts have been devoted to understand the nature of EHE because it is of essential importance to RTP in molecular design⁴⁸⁻⁵⁰. Although the EHE has routinely served as an effective strategy to promote the occurrence of phosphorescence⁵¹, it is still of special interest to explore effective approaches to exert EHE on the design of RTP luminogens.”

“If the counterions were exchanged by heavy halide ions such as iodide (I⁻) and bromide ion (Br⁻), RTP-active organic salts were envisioned to be prepared based on the EHE via anion- π^+ interactions, opening a new avenue to design pure organic RTP luminogens. Moreover, the emission of these molecules would be expected to be manipulated to the desired application in OSMWLEs.”

“Due to the anion- π^+ interactions between the heavy halide ions and the positively-charged aromatic ring, the external heavy halide ions are drawn to be closed to the core chromophore to enhance the SOC to boost the ISC process effectively. Such kind of interactions was defined as heavy-atom-participated anion- π^+ interactions, which could serve as a novel strategy to design RTP luminogens.”

On page 17 in the revised manuscript:

“51. Sun, X., Zhang, B., Li, X., Trindle, C. O. & Zhang, G. External heavy-atom effect via orbital interactions revealed by single-crystal X-ray diffraction. *J. Phys. Chem. A* **120**, 5791–5797 (2016).”

Comment 2. *The absorption spectra in Fig. S13 show that the salts with bromide and iodide counterions are dramatically different vs. the other counterions. The authors explained such*

phenomenon as decreased energy gap due to cation-anion interactions (which do not show up from NMR spectra). This is surprising given that ethanol is strongly solvating. Absorption in other solvents should be checked to make sure that this shift is not due to trivial effects such as the presence of triiodine or tribromine ions.

Our Reply: Thank you very much for your careful and professional suggestion. As you recommended, to deeply explain why the absorption spectra of salts with bromide and iodide counterions are dramatically different with the other counterions, we firstly used starch to check whether it exists triiodine ions based on the chromogenic reaction between starch and triiodine. However, no any chromogenic reaction was observed, verifying no triiodine exists in TPO-I. Then, we measured their absorption spectra in DCM solution. The results show that their absorption spectra are almost unified (Supplementary Fig. 14). Finally, we rechecked their absorption spectra in ethanol solution. All of these organic salts with counterions of I, Br, Cl, F, and PF₆ exhibited the similar absorption spectra as shown in Supplementary Fig. 13. In order to confirm the absorption spectra are correct, we carried out the experiment in another three UV-vis spectrometers, they recorded the same data. Therefore, we speculate that the previous absorption spectra should be wrong due to the problem of UV-vis spectrometer. The related results have been updated in the revised manuscript and Supplementary Information.

On page 4 in the revised manuscript:

“All organic salts have the same absorption peaks at 317 nm accompanying with the weak fluorescence around 440 nm in ethanol solution (Supplementary Fig. 13, 14 and Table 1).”

On pages S9 and S10 in the revised Supplementary Information:

Supplementary Fig. 13 a UV-vis and **b** Photoluminescence (PL) spectra of TPO-I, TPO-Br, TPO-Cl, TPO-F and TPO-P (10 μM) in EtOH solution.

Supplementary Fig. 14 UV-vis spectra of TPO-I, TPO-Br, TPO-Cl, TPO-F and TPO-P (10 μM) in DCM solution.

Comment 3. *In the discussion part, the authors state that “So far, it has been rarely reported to exert the heavy-atom effect on luminogens through non-covalent interactions” is inaccurate, since all external heavy-atom effects occur through non-covalent interactions (see Kasha’s first JPC paper on external-heavy atom effect published in 1953 and many other in 1960s).*

Our Reply: Thank you very much for your kind reminder and professional suggestion. We have revised the part of discussion in the revised manuscript to be more appropriate and reasonable according to the comment and highlighted in red color as follows:

On page 13 in the revised manuscript with highlighted in red color:

“EHE has been used to design RTP luminogens due to the efficient increase of SOC constant between the singlet and triplet states under non-covalent interactions, which boosts the ISC process for efficient phosphorescence. Development of effective approaches for exerting EHE on chromophores to construct RTP luminogens will be of great scientific significance and practical value. Herein, we proposed a unique strategy of heavy-atom-participated anion- π^+ interactions, which was utilized to construct RTP-active organic salts based on TPO derivatives.”

Comment 4. *Regarding the spectroscopic difference between the crystal and film, the authors ascribed it to spectra from different modes of aggregation. This statement requires experimental evidence. For example, the relationship between powder XRD and photoluminescence.*

Our Reply: Thank you very much for your professional suggestion. As you recommended, to illuminate what alters the ratio of the fluorescence and phosphorescence emission intensity and the relationship between powder XRD and photoluminescence, the PL spectra and XRD patterns of TPO-Br microcrystals were measured at different grinding times. With the increase of grinding time, the ratio of phosphorescent/fluorescent intensity and the intensity of XRD peaks gradually decrease. The results revealed that the crystals of TPO-Br were converted into the amorphous powder by mechanical stimulus, which is similar with the phenomenon in a literature (*Angew. Chem. Int. Ed.* 2015, 54, 6270–6273). All the revisions were highlighted in red color in the revised manuscript as follows:

On pages 6-7 in the revised manuscript with highlighted in red color:

“Notably, the crystalline TPO-Br with both fluorescence at 434 nm and RTP at 549 nm is found to respond sensitively to the mechanical grinding. Before grinding, the RTP predominated in the microcrystals or powder of TPO-Br (Supplementary Fig. 27). During grinding, the phosphorescent peak at 549 nm decreased gradually with increasing the grinding time (Supplementary Fig. 28). The above changes in the emission of TPO-Br during the grinding process may be ascribed to the transition from the crystalline to the amorphous form in the solid state, supported by the XRD diffraction. As shown in Supplementary Fig. 29, the diffraction peaks in the XRD patterns suggest the degree of crystallization of microcrystals of TPO-Br decreased in intensity upon grinding. Actually, grinding caused the destruction of rigid surrounding of the crystal state for RTP of TPO-Br, resulting in the decrease of RTP efficiency²⁵. Interestingly, the ratio of intensity between phosphorescence and fluorescence decreased remarkably after grinding. Thus, it may be concluded that the intensity ratio of phosphorescence and fluorescence can be easily tuned by the degree of crystallization. For TPO-I, only phosphorescence was observed before and after grinding although the intensity of XRD peaks became obviously weak (Supplementary Fig. 30 and 31), implying that the fluorescence of TPO-I was seriously quenched due to strong heavy atom effect of iodide.”

On pages S20-S21 in the revised Supplementary Information:

Supplementary Fig. 28 **a** PL spectra and **b** the intensity ratio between phosphorescence (549 nm) and fluorescence (434 nm) for TPO-Br microcrystals under grinding for different time.

Supplementary Fig. 29 XRD patterns of TPO-Br microcrystals under grinding for different time.

Supplementary Fig. 30 PL spectra of TPO-I powder under grinding for different time.

Supplementary Fig. 31 XRD patterns of TPO-I powder before and after grinding for 8 min.

Comment 5. *The authors should list separate lifetime components and weights before adding up for average values in case some important information is missed out.*

Our Reply: Thank you very much for your kind suggestion. According to your suggestion, we summarized the separate lifetime components and weights in the revised Supplementary Information. We have revised the manuscript with a brief discussion about the important information involved in the lifetime data in the revised manuscript with red color highlighted as follows:

On page 6 in the revised manuscript:

“Thus, the yellow emission of TPO-I and TPO-Br is identified to be RTP. **In addition, the lifetimes of TPO-I and TPO-Br at 559 and 549 nm in vacuum were also evaluated with the average values of 52 μ s and 763 μ s, respectively (Supplementary Fig. 18 and Supplementary Table 4), which were slightly larger than that in air, suggesting that such phosphorescence lifetimes of TPO-I and TPO-Br in the solid state were negligibly sensitive to oxygen. All RTP decays can be fitted with two exponents for TPO-I and three exponents for TPO-Br, which is possibly due to the formation of various aggregates in the solid state.**”

On pages S11-S13 in the revised Supplementary Information:

Supplementary Table 1. Fluorescence lifetime data of AIEgens in ethanol at room temperature (298 K; $\lambda_{\text{ex}} = 375$ nm) under air.

	TPO-I	TPO-Br	TPO-Cl	TPO-F	TPO-P
λ_{em} (nm)	447	444	443	438	444
τ_1 (ns) / A_1 (%)	0.46 / 96.43	0.40 / 100	0.39 / 100	0.36 / 100	0.86 / 100
τ_2 (ns) / A_2 (%)	2.61 / 3.57	/	/	/	/
τ (ns) ^a	0.83	0.40	0.39	0.36	0.86

^a τ = average fluorescence lifetime at 434 nm calculated by $\tau = \Sigma A_i \tau_i^2 / \Sigma A_i \tau_i$, where A_i is the pre-exponential factor for lifetime τ_i .

Supplementary Table 2. Fluorescence lifetime data of AIEgens in powder at room temperature (298 K; $\lambda_{\text{ex}} = 375$ nm) under air and in vacuum.

	TPO-Br		TPO-Cl		TPO-F		TPO-P	
	under air	in vacuum	under air	in vacuum	under air	in vacuum	under air	in vacuum
λ_{em} (nm)	434	434	435	435	420	420	422	422
τ_1 (ns) /	0.75 /	0.05 /	0.83 /	0.95 /	0.80 /	0.74 /	1.02 /	0.49 /
A_1 (%)	91.75	92.63	82.62	86.04	100	83.00	100	81.51
τ_2 (ns) /	5.30 /	2.14 /	2.69 /	2.81 /	/	2.37 /	/	1.28 /
A_2 (%)	8.25	7.37	17.38	3.96	/	17.00	/	18.49
τ (ns) ^a	2.52	1.64	1.60	1.55	0.80	1.39	1.02	0.78

^a τ = average fluorescence lifetime at 434 nm calculated by $\tau = \Sigma A_i \tau_i^2 / \Sigma A_i \tau_i$, where A_i is the pre-exponential factor for lifetime τ_i .

Supplementary Table 3. Phosphorescence lifetime data of AIEgens in powder at 77 K under air ($\lambda_{\text{ex}} = 350$ nm).

	TPO-I	TPO-Br
λ_{em} (nm)	559	549
τ_1 (μs) / A_1 (%)	88.16 / 100	1646.75 / 100
τ (μs) ^a	88.16	1646.75

^a τ = average phosphorescence lifetime calculated by $\tau = \Sigma A_i \tau_i^2 / \Sigma A_i \tau_i$, where A_i is the pre-exponential factor for lifetime τ_i .

Supplementary Fig. 18 Time-resolved PL decay of TPO-I (@ 559 nm) and TPO-Br (@ 549 nm) in powder in vacuum.

Supplementary Table 4. Phosphorescence lifetime data of AIEgens in powder at room temperature (298 K; $\lambda_{\text{ex}} = 350$ nm) under air and in vacuum.

	TPO-I		TPO-Br	
	under air	in vacuum	under air	in vacuum
λ_{em} (nm)	559	559	549	549
τ_1 (μs) / A_1 (%)	66.09 / 41.30	9.42 / 2.24	204.00 / 20.14	92.31 / 1.94
τ_2 (μs) / A_2 (%)	27.66 / 58.70	54.65 / 77.76	743.12 / 76.25	388.80 / 4.74
τ_3 (μs) / A_3 (%)	/	/	5.79 / 3.61	858.40 / 63.32
τ (μs) ^a	48.74	52.52	706.42	763.16

^a τ = average phosphorescence lifetime calculated by $\tau = \Sigma A_i \tau_i^2 / \Sigma A_i \tau_i$, where A_i is the pre-exponential factor for lifetime τ_i .

Reviewers' comments:

Reviewer #1 (Remarks to the Author):

Further information on practical usages as well as fundamental optical properties was appropriately supported according to the reviewers' comments. Therefore, I think that this manuscript is suitable for publication as is.

Reviewer #3 (Remarks to the Author):

Tang et al. revised the manuscript based on three main issues raised by the reviewers.

1) A clearer mechanistic explanation of the dual fluorescence-RTP emission from the Br-counterion phosphor in the solid state, i.e., the role of bromide. In order to address this question, the authors went to great length using a combination of experimental and theoretical methods, including concentration-, temperature, solvent- and phase-dependent spectroscopic measurements. I find the conclusions are consistent with the experimentals and calculations and I am inclined to agree with the authors that indeed different modes of aggregation are responsible for the dual emission.

2) The suitability of using the phosphor salt as a coating to produce white light. The authors investigated the coating in working conditions at different temperatures and recorded the prolonged emission spectra in the air. The results suggest that the material is durable enough to have the potential for practical application.

3) The novelty of the current work vs. previous publications. Two of the reviewers questioned the relevance of previous publications on external-heavy atom effect and the authors revised manuscript to cite the original Kasha paper and another one by Zhang et al. The comparison speaks definite novelty and improvement.

Based on the revision of the three most important issues, I thus find the current manuscript readily publishable in Nature Communications.

Reviewer #4 (Remarks to the Author):

This reviewer has looked through the comments of three reviewers and author's response and the revised manuscript. Based on those, this reviewer has concluded that the revised manuscript is still unacceptable in Nat. Commun. with its present form in several aspects as follows:

(1) As reviewer 3 pointed out, the core concept of this work (i.e. external heavy atom effect via the anion- π^+ interaction) has already been reported by G. Zhang et. al (J. Phys. Chem. A 2016, 120, 5791), which greatly reduce the novelty of this work. Although this manuscript has a substantial advance to the previous report (e.g. applications including single-molecule white-light emission), this reviewer is still concerned that this work has enough originality and novelty to be published in top journal like Nat. Commun. Authors should explain what is the real value of this work in the introduction.

(2) Inadequate scientific novelty would be overcome by the demonstration of applications with huge impacts. However, as reviewer 2 pointed out, simple coating of the materials on a light bulb to produce white emission is not very appealing to this reviewer as well. More powerful demonstration will be required.

(3) This reviewer can hardly agree with author's argument on that the heavy halogen atom enhances the CT character of molecules and thus reduces singlet-triplet energy gap. It is well

known that strong CT character greatly reduces the ST energy gap, which is important basis for the design of TADF emitters. However, it is very difficult to accept the generalization of your argument in which the heavy halogen atom can enhance the CT character of molecules; based on your revision, I can see the enhancement of CT character in TPO-I, but this is only one specific example. Author's explain on this problem is still inadequate even in the revised version of the manuscript.

(4) All other issues have been well resolved in the revised manuscript.

(5) What about author's change term "OSMWLEs" to "OWLEs".

In summary, to be published in this journal, authors should improve the scientific novelty and originality (e.g. generalization of single white molecular design) or show high-impact demonstration (e.g. WOLEDs, bio-applications, and so on.).

Responses to the Comments and Suggestions of Reviewers

Reviewer 1:

“Comments: Further information on practical usages as well as fundamental optical properties was appropriately supported according to the reviewers' comments. Therefore, I think that this manuscript is suitable for publication as is.”

Dear Reviewer 1:

We sincerely thank the reviewer's recognition of the novelty, interest and importance of our work.

Reviewer 3:

“Comments: Tang et al. revised the manuscript based on three main issues raised by the reviewers.

1) A clearer mechanistic explanation of the dual fluorescence-RTP emission from the Br-counterionated phosphor in the solid state, i.e., the role of bromide. In order to address this question, the authors went to great length using a combination of experimental and theoretical methods, including concentration-, temperature, solvent- and phase-dependent spectroscopic measurements. I find the conclusions are consistent with the experiments and calculations and I am inclined to agree with the authors that indeed different modes of aggregation are responsible for the dual emission.

2) The suitability of using the phosphor salt as a coating to produce white light. The authors investigated the coating in working conditions at different temperatures and recorded the prolonged emission spectra in the air. The results suggest that the material is durable enough to have the potential for practical application.

3) The novelty of the current work vs. previous publications. Two of the reviewers questioned the relevance of previous publications on external-heavy atom effect and the authors revised manuscript to cite the original Kasha paper and another one by Zhang et al. The comparison speaks definite novelty and improvement.

Based on the revision of the three most important issues, I thus find the current manuscript readily publishable in Nature Communications. ”

Dear Reviewer 3:

We sincerely thank the reviewer's recognition of the novelty, interest and importance of our revised manuscript, and positive evaluation for the improvement of our revised manuscript.

Reviewer 4:

“Comments: This reviewer has looked through the comments of three reviewers and author's response and the revised manuscript. Based on those, this reviewer has concluded that the revised manuscript is still unacceptable in Nat. Commun. with its present form in several aspects as follows:”

Dear Reviewer 4:

We thank you very much for your precious time to review our manuscript and are grateful for your appreciation of our work. We also thank your comments, which help us improve the quality of the manuscript for publication in *Nature Communications*. According to your suggestion, we have carefully carried out the related experiments and revised the manuscript point by point as follows:

***Comment 1.** As reviewer 3 pointed out, the core concept of this work (i.e. external heavy atom effect via the anion- π^+ interaction) has already been reported by G. Zhang et. al (*J. Phys. Chem. A* 2016, 120, 5791), which greatly reduce the novelty of this work. Although this manuscript has a substantial advance to the previous report (e.g. applications including single-molecule white-light emission), this reviewer is still concerned that this work has enough originality and novelty to be published in top journal like *Nat. Commun.* Authors should explain what is the real value of this work in the introduction.*

Our Reply: Thank you very much for the reviewer's kind suggestions. As he/she expressed his/her concern about the originality and novelty of the work, here we explain what is the difference between the previous and this work to illustrate the real value of this work.

1. In Zhang's work, they primarily focused on the specific orbital interactions between the external heavy atom and the luminophore and emphasized the essential importance in design of RTP luminogens (*J. Phys. Chem. A* 2016, 120, 5791), where no aggregation-induced emission (AIE)-related concept was mentioned. However, the work did not propose and discuss the unique

anion- π^+ interactions between the counterions and the aromatic rings, which were firstly proposed to serve as an excellent strategy to construct highly-emissive luminogens with AIE feature by us (J. Am. Chem. Soc. 2017, 139, 16974). Based on our previous work, we are devoted to introducing heavy-atom-participated counterions into the unique anion- π^+ interactions to exert the external heavy atom effect, which was proposed as a new strategy of heavy-atom-participated anion- π^+ interactions for developing RTP luminogens in this work. To our best knowledge, that strategy was not proposed by Zhang et. al, either. In order to further illustrate this novelty of this paper, we have added further statement in the introduction of the revised manuscript with red color highlighted as follows:

On pages 2-3 in the revised manuscript:

“Although the EHE has routinely served as an effective strategy to promote the occurrence of phosphorescence⁵¹, the driving force of interactions between the external heavy atom and the luminophore was not systematically studied. Therefore, in-depth study of the non-covalent interactions is of special interest to explore effective approaches to exert EHE on the design of RTP luminogens with aggregation-induced emission (AIE) feature. Previously, a series of organic salts with AIE features have been developed by our group based on a novel strategy of anion- π^+ interactions⁵²⁻⁵⁶.”

2. Pure organic RTP luminogens with excellent single-molecule white light emission behavior are unusual due to the difficulty of the molecular design. Up to now, only several examples have been published. For example, Chi et. al took advantage of mechanical stimuli to tune the ratio between fluorescence and phosphorescence to realize the white light emission based on a fluorescent-phosphorescent dual-emission compound (Angew. Chem. Int. Ed. 2015, 54, 6270). Tang et. al fully utilized the dual phosphorescence emission behavior of a single organic RTP luminogen to construct single-molecule white light emission (Nat. Commun. 2017, 8, 416). In this work, the single-molecule white light emission based on TPO-Br was successfully obtained by simply tuning the degree of crystallization based on the unique anion- π^+ interactions, which is totally different from the previous reports in design rationale.

3. Compared to the application demonstration in the previous work, the RTP luminogens with single-molecule white light emission in this work have been successfully applied in 3D printing technique to prepare interesting and practical objects such as white light lampshades, which will

be very interesting and largely extend their application.

As a consequence, the strategy of heavy-atom-participated anion- π^+ interactions to design pure organic RTP is of great significance and the application of well-designed RTP luminogens with single-molecule white light emission property in 3D printing is of promising potential, which exhibited the real value of this work.

Comment 2. Inadequate scientific novelty would be overcome by the demonstration of applications with huge impacts. However, as reviewer 2 pointed out, simple coating of the materials on a light bulb to produce white emission is not very appealing to this reviewer as well. More powerful demonstration will be required.

Our Reply: We would like to thank the reviewer very much for his/her kind suggestions. As suggested by the reviewer, the interesting biological application of cell imaging and *in vivo* phosphorescent imaging based on the RTP-active TPO-Br has been demonstrated.

TPO-Br NPs were obtained with the average size of about 250 nm as shown in Supplementary Fig. 39a. Before cell imaging, the cytotoxicity of TPO-Br NPs to live HeLa cells was demonstrated to be very low, which was suggested by the results in Supplementary Fig. 39b. Then, the luminescent imaging of HeLa cells stained by TPO-Br NPs were conducted and the luminescence was monitored by laser scanning confocal microscope. The corresponding confocal images are shown in Supplementary Fig. 39c, where intense yellow emission is clearly observed in HeLa cells, indicating good performance in cell imaging of TPO-Br NPs.

In vivo phosphorescent imaging, due to the average lifetime of TPO-I is as short as 48.7 μ s, it is hard to take the photos with phosphorescent imaging *in vivo* due to the limitation of the IVIS spectrum imaging system. For TPO-Br, its average lifetime is 0.71 ms, whose phosphorescence can be detected by the IVIS spectrum imaging system after UV light excitation is turned off. As shown in Supplementary Fig. 40, we used reported AIEgens TPE-CN and TPO-Br to fabricate fluorescent TPE-CN NPs and phosphorescent TPO-Br NPs with the help of DSPE-PEG, respectively. Fluorescent TPE-CN NPs were used as a reference. In Supplementary Fig. 40a, both the solutions of TPE-CN NPs and TPO-Br NPs showed fluorescence under UV light excitation (*Left*). When the UV light turned off, the solution of TPO-Br NPs exhibited

phosphorescence while no phosphorescence signal was observed in TPE-CN NPs solution (*Right*). Further, we conducted the phosphorescent imaging *in vivo*, as shown in Supplementary Fig. 40b. The fluorescent imaging of living nude mouse after subcutaneous injection of TPE-CN NPs suffered from the serious autofluorescence stemmed from biological background. In sharp contrast, the living nude mouse after injection of TPO-Br NPs was clearly observed the phosphorescence signals with extremely low background due to the longer lifetime of phosphorescence than that of fluorescence. Thus the phosphorescence properties of TPO-Br make it suitable for imaging with high signal to noise *in vivo*. We have added further discussion in the revised manuscript and revised supplementary information with red color highlighted as follows:

On page 14 in the revised manuscript:

“Thus, heavy-atom-participated anion- π^+ interactions play a crucial role in the emergence of RTP for TPO-I and TPO-Br. **It is worth to mention that the RTP properties of TPO-Br with the longer lifetime makes it well suitable for cell imaging and *in vivo* phosphorescent imaging with extremely low autofluorescence compared to fluorescent probes (Supplementary Figs. 39 and 40).**”

On page S28-S31 in the revised supplementary Information

Supplementary Fig. 39 **a** DLS analysis of DSPE-PEG-encapsulated TPO-Br NPs. **b** Cell viabilities of HeLa cells in the presence of different concentrations of TPO-Br NPs. **c** Bright-field, luminescent and merged images of HeLa cells stained with TPO-Br NPs (1 mg/mL) based on laser scanning confocal microscope (LSCM). Excitation wavelength: 405 nm, emission was acquired in the range from 500 to 720 nm.

Supplementary Fig. 40 *In vivo* phosphorescent imaging. **a** Luminescent photos of the aqueous solutions of DSPE-PEG-encapsulated TPE-CN NPs and TPO-Br NPs (*Left*: fluorescence; *Right*: phosphorescence). Blank is the DSPE-PEG solution. **b** Luminescent imaging in living mice after subcutaneous injection of TPE-CN NPs (fluorescence) and TPO-Br NPs (Phosphorescence).

TPO-Br NPs were obtained with the average size of about 250 nm as shown in Supplementary Fig. 39a. Before cell imaging, the cytotoxicity of TPO-Br NPs to live HeLa cells was demonstrated to be very low, which was suggested by the results in Supplementary Fig. 39b. Then, the luminescent imaging of HeLa cells stained by TPO-Br NPs were conducted and the luminescence was monitored by laser scanning confocal microscope. The corresponding confocal images are shown in Supplementary Fig. 39c, where intense yellow emission is clearly observed in HeLa cells, indicating good performance in cell imaging of TPO-Br NPs.

Due to the average lifetime of TPO-Br is as long as 0.71 ms, its phosphorescence can be detected by the IVIS spectrum imaging system after UV light excitation is turned off. As shown in Supplementary Fig. 40, reported AIEgens TPE-CN and TPO-Br were utilized to fabricate

fluorescent TPE-CN NPs and phosphorescent TPO-Br NPs with the help of DSPE-PEG, respectively. In Supplementary Fig. 40a, both the solutions of TPE-CN NPs and TPO-Br NPs showed fluorescence under UV light excitation (*Left*). When the UV light turned off, the solution of TPO-Br NPs exhibited phosphorescence while no phosphorescence signal was observed in TPE-CN NPs solution (*Right*). Further, we conducted the phosphorescent imaging *in vivo*, as shown in Supplementary Fig. 40b. The fluorescent imaging of living nude mouse after subcutaneous injection of TPE-CN NPs suffered from the serious autofluorescence stemmed from biological background. In sharp contrast, the living nude mouse after injection of TPO-Br NPs was clearly observed the phosphorescence signals with extremely low background due to the longer lifetime of phosphorescence than that of fluorescence. Thus the phosphorescence properties of TPO-Br make it suitable for imaging with high signal to noise *in vivo*.

Comment 3. This reviewer can hardly agree with author's argument on that the heavy halogen atom enhances the CT character of molecules and thus reduces singlet-triplet energy gap. It is well known that strong CT character greatly reduces the ST energy gap, which is important basis for the design of TADF emitters. However, it is very difficult to accept the generalization of your argument in which the heavy halogen atom can enhance the CT character of molecules; based on your revision, I can see the enhancement of CT character in TPO-I, but this is only one specific example. Author's explain on this problem is still inadequate even in the revised version of the manuscript.

Our Reply: Thank you very much for your professional and kind suggestions. Firstly, we agree with the reviewer's viewpoint that the heavy halogen atom cannot enhance the CT character of molecules. Actually, the statements involved CT character in the revised manuscript did not referred to the argument on that the heavy halogen atom enhances the CT character of molecules and thus reduces singlet-triplet energy gap. For example:

In the first paragraph of page 9 in the revised manuscript, "The introduction of bromine and iodine firstly enhances SOC in TPO-Br and TPO-I owing to the heavy atom effect. Meanwhile, it was also revealed that there displays remarkable charge transfer character from the bromine/iodine to π -conjugated TPO core⁵¹". We just want to express that remarkable charge

transfer character from the bromine/iodine to π -conjugated TPO core were observed for TPO-Br and TPO-I.

In the second paragraph of page 9 in the revised manuscript, “From T₁, T₂, T₃ to T₄, the bromine component of the hole NTO increases from 29.0%, 51.4%, 93.3% to 95.5%, indicating different heavy atom effect on SOC, the degree of n- π^* / π - π^* mixture and charge transfer characters in the four triplet states.”. We stated the effect of heavy atom on SOC, the degree of n- π^* / π - π^* mixture and charge transfer characters in the four triplet states of TPO-Br.

In the third paragraph of page 9 in the revised manuscript, “For TPO-I, the contributions of iodine to the excited states in TPO-I are far larger than those of bromine in TPO-Br, such as more outstanding heavy atom effect and charge transfer characters.”. We want to illustrate that the contributions of iodine to the excited states lead to strong charge transfer characters of TPO-I.

Thus, in order to understand clearly, we have modified the related statements as follows:

On page 9 in the revised manuscript:

“The introduction of bromine and iodine enhances SOC in TPO-Br and TPO-I owing to the heavy atom effect. Meanwhile, remarkable charge transfer character from the bromine/iodine to π -conjugated TPO core was also observed⁵¹.”

“From T₁, T₂, T₃ to T₄, the bromine component of the hole NTO increases from 29.0%, 51.4%, 93.3% to 95.5%, indicating the effect of different heavy atoms on SOC, the degree of n- π^* / π - π^* mixture and charge transfer characters in the four triplet states.”

“For TPO-I, the contributions of iodine to the excited states in TPO-I are far larger than those of bromine in TPO-Br, such as more outstanding heavy atom effect,”

Comment 4. All other issues have been well resolved in the revised manuscript.

Our Reply: We sincerely thank the reviewer’s recognition for the improvement of our revised manuscript.

Comment 5. What about author’s change term “OSMWLEs” to “OWLEs”.

Our Reply: We would like to thank you very much for your careful reminder. However, we are very sorry that we want to keep the term of “OSMWLEs”, which is the abbreviation of organic single-molecule white light emitters. As we known, organic white light emitters (OWLEs) have been reported usually due to the tireless efforts of scientists. Among them many OWLEs are composed of multicomponents with three (blue, green and red) or two (blue and yellow) emission, which more or less suffered from the phase segregation, color aging, poor reproducibility and stability, as well as laborious preparation. OSMWLEs based on organic single molecule could perform better than OWLEs and have been increasingly pursued by researchers. Although recent research demonstrated that highly-efficient OSMWLEs could be realized by the combination of fluorescent and phosphorescent or two different phosphorescent emission of pure organic RTP luminophores, the research on this field is still in its infancy and remains challenging due to the lack of excellent pure organic RTP luminogens. Therefore, it is of great significance to develop pure organic RTP luminogens with two (blue and yellow) or three (blue, green, and red) complementary emission colors for OSMWLEs. We are devoted to developing excellent RTP luminogens to realize OSMWLEs in this work, thus we keep the term of “OSMWLEs”.

REVIEWERS' COMMENTS:

Reviewer #4 (Remarks to the Author):

Although this reviewer still has some concerns about the novelty, bio-imaging application is quite appealing to me. Thus, the current form of the manuscript is publishable.

-As for the novelty of this work, this reviewer think that simple combination of AIE and heavy-atom effect is not enough novel to publish in Nat. Commun. AIE is currently very well-known concept and thus not novel anymore. The heavy atom effect by counter ions is also not enough novel due to the existence of J. Phy. Chem. paper which I have mentioned before.

-All other issues are now well addressed.

Responses to the Comments and Suggestions of Reviewers

Reviewer 4:

“Comments: Although this reviewer still has some concerns about the novelty, bio-imaging application is quite appealing to me. Thus, the current form of the manuscript is publishable.”

Dear Reviewer 4:

We sincerely thank the reviewer's recognition and interest of bio-imaging application of our revised manuscript. We also thank the reviewer's agreement to publishing in the current form of the manuscript.

Comment 1. As for the novelty of this work, this reviewer think that simple combination of AIE and heavy-atom effect is not enough novel to publish in Nat. Commun. AIE is currently very well-known concept and thus not novel anymore. The heavy atom effect by counter ions is also not enough novel due to the existence of J. Phy. Chem. paper which I have mentioned before.

Our Reply: We thank you very much for your precious time to review our manuscript and help us improve the quality of the manuscript for publication in *Nature Communications*. As the reviewer still has some concerns about the novelty of the work, here it is necessary to explain the novelty of this work to address the concern of the reviewer.

As the reviewer mentioned that AIE is currently very well-known concept, research on luminogens with AIE feature (AIEgens) should be very interesting and important in various area. Although many AIEgens have been developed, it still remains huge challenge, especially in the in-depth mechanism research of AIE and design of novel AIEgens with unique function for wide applications such as room temperature phosphorescence and organic single-molecule white light emission. In this work, we introduce the unique anion- π^+ interactions and heavy atom effect to construct AIE-active RTP materials with single-molecule white light emission. It is interesting that the marriage between anion- π^+ interactions and heavy atom effect enable the external heavy atom effect (EHE) effectively. This work is not just a simple combination of AIE and heavy-atom effect. The novel concept of heavy-atom-participated anion- π^+ interactions as a driving force to exert EHE on designing RTP materials was not proposed by Prof. Zhang in the paper of *J. Phy. Chem.* and others.

Besides, organic single-molecule white light emission and their 3D printing application are also very interesting and novel in this work, because pure organic RTP luminogens with excellent single-molecule white light emission properties are indeed unusual and difficult to be designed. Up to now, only several examples have been published. In this work, the single-molecule white light emission based on TPO-Br was successfully obtained by simply tuning the degree of crystallization based on the unique anion- π^+ interactions, which is totally different from the previous reports in design rationale. Moreover, the RTP luminogens with single-molecule white light emission in this work have been successfully applied in 3D printing technique to prepare interesting and practical objects, which could not be realized by the previously-reported RTP luminogens with white light emission. Thus it largely extend their application.

It should be emphasized that this reviewer indeed help us explore the interesting biological application involving cell imaging and *in vivo* phosphorescent imaging based on the RTP-active TPO-Br. Compared to their fluorescent counterparts, TPO-Br with phosphorescence properties exhibited superior signal to noise *in vivo*, avoiding the serious autofluorescence stemmed from biological background. There is no doubt that this biological application is another important novelty of this work.

As a consequence, the strategy of **heavy-atom-participated anion- π^+ interactions** to design **pure organic RTP** is of great significance and the application of well-designed RTP luminogens with **single-molecule white light emission** property in **3D printing** and **bio-imaging** is of promising potential, which exhibited the novelty of this work.

In order to further highlight the novelty of this paper, we have added further statement in the introduction of the revised manuscript with red color highlighted as follows:

On pages 2-3 in the revised manuscript:

“Although the EHE has routinely served as an effective strategy to promote the occurrence of phosphorescence⁵¹, the driving force of interactions, **especially non-covalent interactions such as ion- π interactions**, between the external heavy atom and the luminophore **has not been** systematically studied.”

Comment 2. All other issues are now well addressed.

Our Reply: We sincerely thank the reviewer's recognition of our revised manuscript.